# Self-Supervised Flow Matching for Scalable Multi-Modal Synthesis

**Hila Chefer** [* 1]  **Patrick Esser** [* 1]

**Dominik Lorenz** [1]  **Dustin Podell** [1]  **Vikash Raja** [1]  **Vinh Tong** [1]  **Antonio Torralba** [1 2]  **Robin Rombach** [1]

https://bfl.ai/research/self-flow

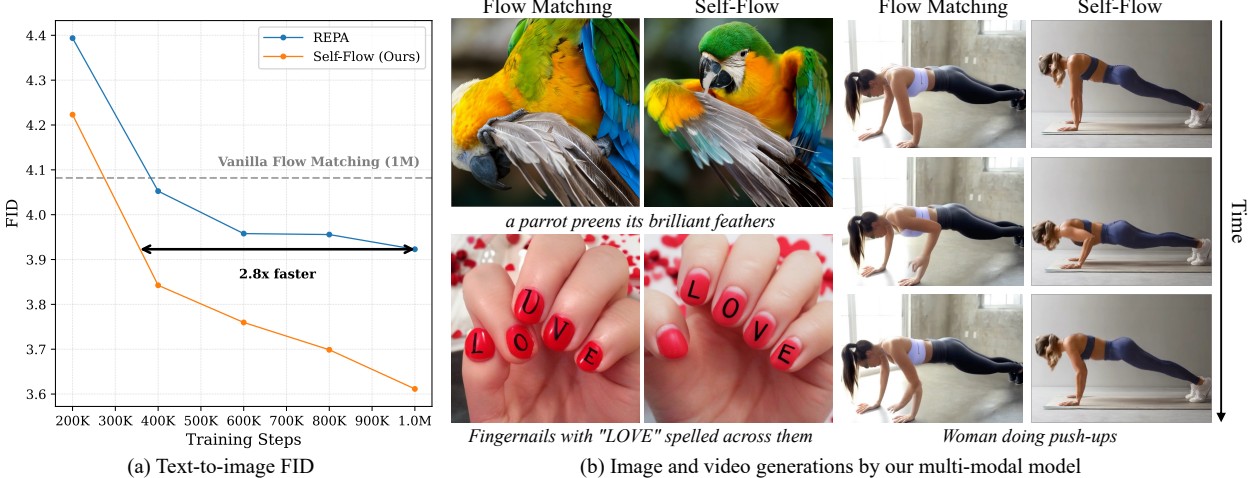

Flow Matching    Self-Flow      Flow Matching    Self-Flow

*a parrot preens its brilliant feathers*

*Fingernails with "LOVE" spelled across them*

*Woman doing push-ups*

(a) Text-to-image FID        (b) Image and video generations by our multi-modal model

*Figure 1.* **Results by *Self-Flow*, our self-supervised flow matching framework.** (a) On image generation, our method converges $2.8\times$ faster than REPA (Yu et al., 2024) *without any external models.* Notably, REPA plateaus while our method continues to improve. (b) Compared to flow matching, our approach improves structural coherence, text rendering, and temporal consistency.

## Abstract

Strong semantic representations improve the convergence and generation quality of diffusion and flow models. Existing approaches largely rely on external models, which require separate training, operate on misaligned objectives, and exhibit unexpected scaling behavior. We argue that this dependence arises from the model's training objective, which poses a denoising task with little incentive to learn semantic representations. We introduce *Self-Flow*: a self-supervised flow matching paradigm that integrates representation learning within the generative framework. Our key mechanism, *Dual-Timestep Scheduling*, applies heterogeneous noise levels across tokens, creating an information asymmetry that forces the model to infer missing information from corrupted inputs. This drives learning strong representations alongside generative capabilities without external supervision. Our method generalizes across modalities and enables multi-modal training while following expected scaling laws, achieving superior image, video, and audio generation.

## 1. Introduction

Modern generative models (Rombach et al., 2022a; Peebles & Xie, 2023; Ma et al., 2024; Esser et al., 2024), trained on vast data using extensive computational resources, can be dramatically improved by aligning their internal features with those of a frozen image encoder, for example, the 86M parameter model *DINO* (Yu et al., 2024). This encoder was trained not to generate, but to *discriminate*, i.e., to cluster images by semantic similarity. Its effectiveness for generative modeling exposes a gap: flow models do not learn strong representations on their own, although they help generation. External alignment offers a practical remedy: borrowing representations from a model that *did* learn them.

However, this approach has fundamental limitations. First, external alignment fails to uphold expected scaling laws, with stronger encoders often exhibiting diminished or even negative returns (Singh et al., 2025) (Sec. 3.2). Moreover, as we demonstrate in this work (see Sec. 4), scaling the generative model does not yield proportional improvements when relying on external alignment. Second, these methods fail to generalize across modalities: for video and audio generation, we find that alignment with most external encoders actually harms performance (Sec. 4), making external alignment less suitable for multi-modal models that must handle diverse data distributions within a single framework. Finally, it is difficult to anticipate which encoder will be effective for

---

[1]Black Forest Labs [2]MIT. [*]Equal contribution. Correspondence to: Hila Chefer <hila@blackforestlabs.ai>, Patrick Esser <patrick@blackforestlabs.ai>.

*Proceedings of the $43^{rd}$ International Conference on Machine Learning*, Seoul, South Korea. PMLR 306, 2026. Copyright 2026 by the author(s).

a given task. For example, aligning text-to-image models with SigLIP 2 (Tschannen et al., 2025) performs worse than DINOv2 (Oquab et al., 2024) (Sec. 4), despite the former being explicitly trained with text supervision and multi-aspect ratio support, properties seemingly better suited for the task.

To avoid the use of external representations, existing approaches opt to rely on the model's natively learned features and the semantic asymmetry between different layers (Jiang et al., 2025; Haghighi et al., 2025). However, such formulations remain limited by the semantics naturally learned by the flow objective, and lag behind external alignment.

In contrast to both approaches above, we propose to directly integrate a self-supervised framework into flow matching to actively strengthen representations beyond those learned by the generative objective alone. To this end, we propose *Dual-Timestep Scheduling*, which applies two distinct noise levels to different subsets of input tokens, creating an information asymmetry in which some tokens are more heavily corrupted than others. We perform two forward passes: one with the mixed, heterogeneously-noised input, and one with a cleaner input where all tokens are noised at the lower of the two levels. The self-supervised objective is to predict, from the mixed input, the representations the model produces given the cleaner input. Combined with the standard flow loss on the heterogeneously-noised input, the model thus learns both dense, flow-based reconstruction and semantic feature prediction within a unified framework.

Since our formulation operates purely on the model's internal representations without relying on external encoders, it naturally extends to both single-modality and joint multi-modal training. Fig. 1 shows qualitative examples from our jointly-trained image, video, and audio model. Compared to standard flow matching, our method yields improvements in structural coherence, particularly for challenging structures like faces and hands, as well as text rendering accuracy and temporal consistency in video. Moreover, our method is agnostic to autoencoder choice: we demonstrate consistent improvements across SD (Rombach et al., 2022a), FLUX.2 (Black Forest Labs, 2025), Wan2.2 (Wan et al., 2025), Songbloom (Yang et al., 2025), and representation autoencoders (Zheng et al., 2025) (Sec. 4).

We evaluate on image, video, audio, and multi-modal generation, and show that our framework outperforms leading external alignment methods in each setting. These results suggest that joint optimization of generation and representations offers a robust, scalable, and general path forward.

**Conflict of Interest Disclosure**  The authors HC, PE, DL, DP, VR, VT, AT, RR are employed by or are advisors to Black Forest Labs, which leads the development of FLUX, which was one of the models evaluated in the paper.

## 2. Related Work

**Representation Learning.** Representation learning aims to learn powerful semantics through diverse pretraining objectives. Contrastive methods such as SimCLR (Chen et al., 2020a), MoCo (He et al., 2019; Chen et al., 2020b; 2021), and BYOL (Grill et al., 2020) learn maximize agreement between augmented views of the same image. CLIP (Radford et al., 2021) and SigLIP (Zhai et al., 2023; Tschannen et al., 2025) extend this paradigm to vision-language alignment, enabling zero-shot transfer across tasks. DINO variants (Caron et al., 2021; Oquab et al., 2024; Siméoni et al., 2025) train a student network to match a momentum-updated teacher. Masked autoencoding (MAE) (Vincent et al., 2010; Pathak et al., 2016) and its variants (He et al., 2022; Bao et al., 2021; Assran et al., 2023) learn representations by reconstructing masked portions of the input.

**Representation Alignment for Generation.** Existing methods can be categorized into those that align with external models and those that do not. Recent work has shown that aligning diffusion and flow model features with external pretrained encoders can significantly accelerate training and improve generation quality (Yu et al., 2024; Yao & Wang, 2025; Leng et al., 2025a;b; Pernias et al., 2023), with extensions to domain-specific settings such as physics in video (Zhang et al., 2025) and geometry in 3D (Wu et al., 2025b). Another line of work trains generative models directly on semantic representations rather than reconstruction-driven latents (Wu et al., 2025a; Zheng et al., 2025), though these methods remain tied to specific encoders which limits their reconstruction performance and hinders adaption across resolutions and modalities. For completeness, we show in Sec. 4 and App. H.2 that our method improves the performance of RAE (Zheng et al., 2025), demonstrating our method's robustness to autoencoder choice. As discussed in Sec. 1, external alignment methods exhibit fundamental limitations such as unexpected scaling behavior (Sec. 3.2) and limited generalization across datasets and modalities (Sec. 4), motivating the need for a unified approach that eliminates dependence on external models entirely.

Existing methods that do not rely on external models can be broadly categorized into two groups. The first incorporates explicit self-supervised objectives at the cost of modifying the model's training dynamics, often necessitating an additional stage of pure diffusion fine-tuning to close the train-inference gap (Zheng et al., 2023; Gao et al., 2023; Chen et al., 2025b; Zhu et al., 2024). The second preserves the diffusion framework (Jiang et al., 2025; Wang & He, 2025; Haghighi et al., 2025; Hu et al., 2023) and employs diffusion features to perform alignment or guidance. However, the latter methods rely on the assumption that deeper layers naturally learn strong representations, while the former obtain results that are less favorable (Wang & He, 2025;

Zheng et al., 2023; Gao et al., 2023). Overall, methods that use external representations have consistently outperformed those without. Our novel *Self-Flow* approach closes this gap: by integrating self-supervised learning directly into flow matching, we surpass external alignment methods without requiring any external models.

# 3. Method

## 3.1. Preliminaries

Flow matching models learn to transport samples from a simple noise distribution to the data distribution by modeling a continuous-time probability path. Our approach builds on rectified flows (Liu et al., 2022; Albergo & Vanden-Eijnden, 2023; Lipman et al.), which constructs straight-line trajectories between noise and data.

Let $\mathbf{x}_0 \in \mathbb{R}^{N \times C}$ denote clean data represented as a sequence of $N$ tokens, each of dimension $C$. This formulation naturally accommodates diverse input modalities, including image patches, video frames, and audio segments. We define a probability path by linearly interpolating between a noise distribution $\mathbf{x}_1 \sim \mathcal{N}(\mathbf{0}, \mathbf{I})$ and the data distribution:

$$\mathbf{x}_t = (1 - t)\mathbf{x}_0 + t\mathbf{x}_1, \quad t \in [0, 1], \qquad (1)$$

where $t$ parameterizes the interpolation, with $t = 1$ corresponding to pure noise and $t = 0$ corresponding to clean data. The velocity field along this path is given by $\mathbf{v}_t = \frac{d\mathbf{x}_t}{dt} = \mathbf{x}_1 - \mathbf{x}_0$. A neural network $f_\theta(\mathbf{x}_t, t)$ is trained to predict the velocity field by minimizing:

$$\mathcal{L}_{\text{gen}} = \mathbb{E}_{\mathbf{x}_0, \mathbf{x}_1, t}\|f_\theta(\mathbf{x}_t, t) - (\mathbf{x}_1 - \mathbf{x}_0)\|^2, \qquad (2)$$

where $t \sim p(t)$ denotes the timestep sampling distribution (see App. B.2). At inference, generation proceeds by solving the ordinary differential equation (ODE) $\frac{d\mathbf{x}_t}{dt} = f_\theta(\mathbf{x}_t, t)$ from pure noise $\mathbf{x}_1 \sim \mathcal{N}(\mathbf{0}, \mathbf{I})$ backwards in time to $t = 0$, yielding a sample from the learned data distribution.

Recent works (Yu et al., 2024; Leng et al., 2025a; Jiang et al., 2025) demonstrate that flow matching training benefits substantially from feature alignment. Given representations from a teacher model $r_\phi$, these methods augment training with an auxiliary objective:

$$\mathcal{L}_{\text{rep}} = -\mathbb{E}_{\mathbf{x}_0, \mathbf{x}_1, t, t'} \, \text{sim}\left(h_\theta^{(l)}(\mathbf{x}_t, t), r_\phi^{(k)}(\mathbf{x}_{t'}, t')\right), \quad (3)$$

where sim denotes a similarity metric, $l$, $k$ denote layer indices of the flow model and teacher, respectively, and $h_\theta^{(l)}(\mathbf{x}_t, t) = h_\theta(f_\theta^{(l)}(\mathbf{x}_t, t))$ is an MLP projection head. This loss aligns the features at the $l$-th layer of the flow model $f_\theta$ with the representation features $r_\phi^{(k)}(\mathbf{x}_{t'}, t')$ from layer $k$. Alignment methods achieve optimal performance when aligning with an external pretrained encoder, with

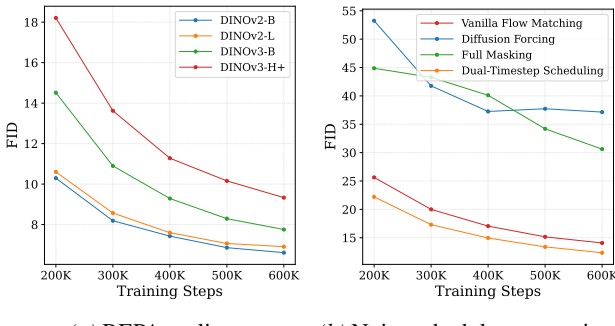

*(a)* REPA scaling     *(b)* Noise scheduler comparison

*Figure 2.* **Motivation.** (a) Scaling DINO (v2-B< v2-L< v3-H+) paradoxically degrades the generations quality using REPA. (b) Diffusion forcing and full masking create a train-inference gap which degrades generations. Our Dual-Timestep Scheduling improves generation even without a self-supervised objective.

DINOv2-B (Oquab et al., 2024) being the predominant choice (Yu et al., 2024; Leng et al., 2025a; Singh et al., 2025). However, existing evaluations focus primarily on class-conditional ImageNet generation (Deng et al., 2009), a dataset heavily represented in DINOv2 training, potentially biasing the reported results. In this work, we demonstrate that reliance on external models leads to unexpected behavior across data distributions, model scales, and modalities.

## 3.2. Motivation

We begin by presenting experiments that motivate our unified framework by testing the scaling laws of external alignment methods. Following the setup of REPA (Yu et al., 2024) (Eq. 3), we replace the DINOv2-B backbone with increasingly stronger variants (DINOv2-L, DINOv3-B, DINOv3-H+). Figure 2a reveals an inverse correlation: stronger representation learners consistently degrade generation quality. DINOv2-B, the smallest and weakest variant, achieves the best FID, while the most capable model, DINOv3-H+, performs the worst. This suggests that external alignment creates a bottleneck: the generative model becomes dependent on a fixed external representation that may not align with the generative goal. Instead of relying on fixed representations, we want to strengthen them within the generative framework itself.

## 3.3. Dual-Timestep Scheduling

In standard flow matching, uniform noise is applied to all tokens, resulting in a denoising task that can often be solved by local correlations alone. To encourage the learning of stronger, more global representations across the model, we introduce information asymmetry: by applying different noise levels to different tokens, the model is encouraged to use cleaner tokens to infer noisy tokens. The key challenge is how to introduce such heterogeneous noise without disrupting the underlying generative dynamics.

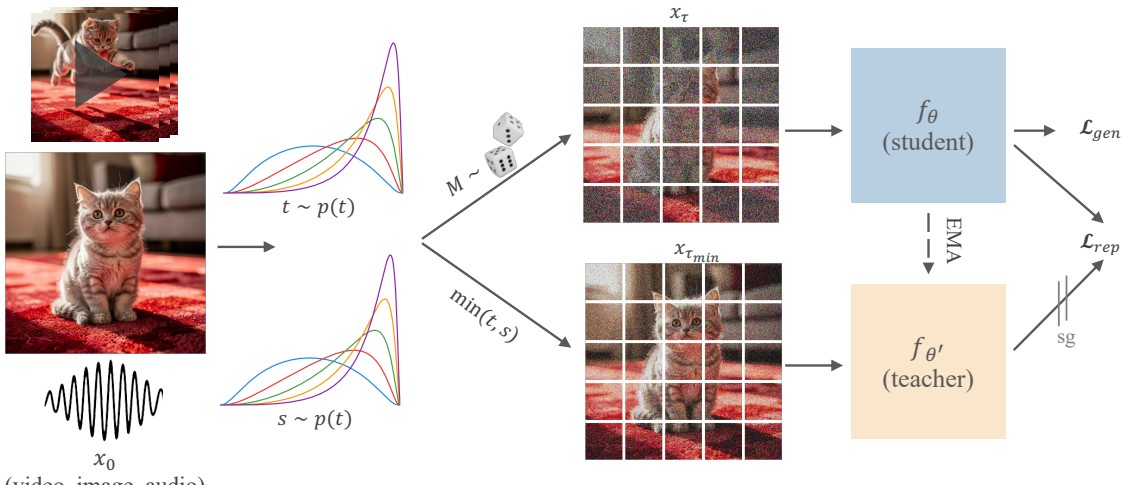

*Figure 3.* **Illustration of our method.** Given a clean input $x_0$, we draw two timesteps $t, s$, and a random mask $M$, then noise each token according to its assigned timestep. The teacher input is noised with $\tau_{min} = \min\{t, s\}$, creating an information asymmetry compared to the student. The student is trained to simultaneously denoise the input and reconstruct the teacher's features given its mixed-noised view.

One intuitive strategy is to randomly set $t = 1$ for a subset of tokens, fully masking them. Another is to sample an independent noise level for each token, similar to diffusion forcing (Chen et al., 2024). In Fig. 2b, we compare these approaches with flow matching and our proposed scheduling method. Both naive masking and diffusion forcing substantially degrade the generation quality. We attribute this to a train–inference gap: during inference, the model must denoise uniformly noised inputs at both low and high noise levels, a regime that is rarely encountered during training.

To address this mismatch, we propose Dual-Timestep Scheduling (DTS). The core idea is to sample *two* timesteps from the noise distribution (Fig. 3). The higher of the two noises effectively corrupts information, while the cleaner one serves as context. Specifically, given an input $x_0$ we:

1. Sample two timesteps: $t, s \sim p(t)$

2. Sample a mask $M = \{i \in \{1, \ldots, N\} \mid u_i < \mathcal{R}_M\}$, with $u_i \overset{\text{iid}}{\sim} \mathcal{U}(0, 1)$ and a masking ratio $\mathcal{R}_M \leq 0.5$.

3. Construct a Dual-Timestep $\boldsymbol{\tau} \in \mathbb{R}^N$ to noise $\mathbf{x}_0$:

$$\tau^i = \begin{cases} s & \text{if } i \in M \\ t & \text{otherwise} \end{cases} \quad (4)$$

$$\mathbf{x}_\tau = \text{diag}(\mathbf{1} - \boldsymbol{\tau})\mathbf{x}_0 + \text{diag}(\boldsymbol{\tau})\mathbf{x}_1 \quad (5)$$

This approach strikes a balance between homogeneous noising, which fails to encourage strong global relations, and the fully heterogeneous approach which fails to simulate inference behavior during training, while maintaining the marginal timestep distribution per token. A theoretical discussion on DTS can be found in App. A.

Interestingly, as observed in Fig. 2b, DTS alone, applied to the vanilla flow matching training, is able to slightly improve the generation quality even without an explicit self-supervised objective. Intuitively, this can be attributed to the presence of cleaner information in the input, which helps the model perform the denoising task, thus the model is implicitly encouraged to consider global relations, which in turn improves its generative capabilities.

### 3.4. Self-Flow

Next, we show how to leverage the information asymmetry created by DTS to encourage the model to learn stronger representations. As illustrated in Fig. 3, we maintain two models: a student network $f_\theta$ that learns from heterogeneously noised inputs $\mathbf{x}_\tau$, and an EMA teacher network $f_{\theta'}$ that has the advantage of observing the cleaner $\mathbf{x}_{\tau_{min}}$ which is noised by $\tau_{min} = \min(\boldsymbol{\tau}) \in \{t, s\}$. Based on this setup, we can now devise a feature alignment loss where the student learns to reconstruct the teacher's features from its partial, corrupt view of the input. Formally, our representation alignment objective is given by using the teacher network $f_{\theta'}$ as the representation network and integrating the dual timestep $\boldsymbol{\tau}$ into Eq. 3, using cosine similarity as the alignment metric:

$$\mathcal{L}_{\text{rep}} = -\mathbb{E}_{\mathbf{x}_0, \mathbf{x}_1, \boldsymbol{\tau}} \cos\left(h_\theta^{(l)}(\mathbf{x}_{\boldsymbol{\tau}}, \boldsymbol{\tau}), f_{\theta'}^{(k)}(\mathbf{x}_{\tau_{min}}, \tau_{min})\right) \quad (6)$$

Following insights from Yu et al. (2024) on the evolution of semantics in diffusion models, we choose $l < k$.

To perform this task, the student is encouraged to actively leverage the cleaner tokens to infer the representations for the noisier tokens, forming global connections that transcend simple locality. Our training objective combines generation and representation learning, parametrized by a scaling factor $\gamma$:

$$\mathcal{L} = \mathcal{L}_{\text{gen}} + \gamma \cdot \mathcal{L}_{\text{rep}}. \quad (7)$$

*Table 1.* **Quantitative results on ImageNet 256×256.** Our method outperforms REPA, despite its use of DINOv2 which heavily relies on ImageNet for training. Our method is beneficial in combination with representation autoencoders.

| Model | Steps | FID↓ | sFID↓ | IS↑ | Pre.↑ | Rec.↑ |
|---|---|---|---|---|---|---|
| *Without external representations* | | | | | | |
| SiT-XL/2 | 7M | 8.3 | 6.30 | 130.57 | 0.69 | 0.67 |
| SRA | 4M | 7.27 | 5.87 | 143.06 | 0.69 | 0.68 |
| **Ours** | 4M | **5.70** | **4.97** | 151.40 | **0.72** | 0.67 |
| *With external representations* | | | | | | |
| REPA | 4M | 5.89 | 5.73 | **157.66** | 0.70 | **0.69** |
| *With representation autoencoders* | | | | | | |
| RAE | 1M | 3.24 | 6.73 | 218.53 | 0.83 | 0.54 |
| RAE + Ours | 1M | **2.95** | **5.50** | **222.34** | **0.84** | **0.56** |

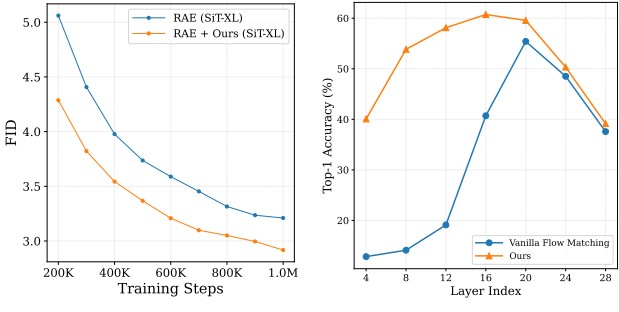

*(a)* Our method w/ RAE     *(b)* Linear probing on ImageNet

*Figure 4.* **Autoencoder generalization and improved representations.** (a) Our method improves training and generation in RAE (Zheng et al., 2025), demonstrating compatibility with semantic latent spaces. (b) Linear probing confirms that our method learns stronger representations than standard flow matching.

## 4. Experiments

We evaluate our method on class-to-image (ImageNet), text-to-image (T2I), text-to-video (T2V), text-to-audio (T2A), and multi-modal generation. Through quantitative results, qualitative comparisons, and scaling experiments, we demonstrate the effectiveness of our approach, its adaptability across tasks and modalities, and its scaling properties.

**Implementation Details.** For ImageNet experiments, we use SiT-XL (Ma et al., 2024) with the REPA setup. All other experiments use the FLUX.2 (Black Forest Labs, 2025) transformer with domain-specific autoencoders. Unless noted, all models are ∼625M parameters. For text-to-image (T2I), we use the Stable Diffusion autoencoder (following the ImageNet setup) and train on 20M text-image pairs; for text-to-video (T2V), the Wan2.2 (Wan et al., 2025) autoencoder with 6M videos; for text-to-audio (T2A), the Songbloom autoencoder (Yang et al., 2025) with FMA (Defferrard et al., 2017). All evaluations are conducted on holdout sets of the corresponding training sets. See App. B for further training and sampling details.

### 4.1. Single Modality Experiments

**Baselines.** We compare against vanilla flow matching and the leading methods from both categories of representation alignment: with and without external models. For each category, we select the best performing approach applicable across all tested modalities: REPA (Yu et al., 2024) for external alignment and SRA (Jiang et al., 2025) for methods without external models (see App. B.5). To ensure a thorough comparison, we additionally evaluate external encoders that are, in theory, better suited than DINO to each task: SigLIP 2 (Tschannen et al., 2025) for text-to-image (trained with text supervision and multi-aspect ratio support), V-JEPA 2 (Bardes et al., 2024) and Depth Anything 3 (Lin et al., 2025) for video, and MERT (Li et al., 2024c) for audio.

**Quantitative Results.** On ImageNet (Tab. 1), our method outperforms REPA (FID 5.70 vs 5.89) without external representations and despite REPA using DINOv2, itself heavily trained on ImageNet. To our knowledge, we are the first to show self-supervised learning outperforming external alignment on ImageNet. To demonstrate generalization to arbitrary latent spaces, we apply our method to RAE (Zheng et al., 2025), in the same setup as the other experiments. As shown in Fig. 4a and Tab. 1, this yields significant improvements (FID 3.24 → 2.95). See App. H.2 for further details. Finally, Fig. 4b shows linear probing results after 2M training steps. Our method significantly boosts the representation quality of early and mid layers, confirming that representations improve alongside generations.

On the T2I task (Fig. 5a,b, Tab. 2), our method achieves the best FID (3.61) among all methods, outperforming both external alignment approaches (REPA: 3.92, SigLIP 2: 3.97) and methods without external models (SRA: 3.70). Notably, we outperform REPA even when computing the Fréchet distance score with DINOv2 features (Fig. 5b, 167.98 vs 173.35) despite REPA explicitly aligning with DINOv2 features, a gap no other baseline closes. Our method also achieves the highest CLIP score, indicating superior text-image alignment.

Our approach shows particularly strong gains on *video generation* (Fig. 5c, Tab. 3), achieving the best FVD (47.81) and FID (8.92, per frame) by a significant margin; the next best method (REPA) trails by nearly 2 FVD points. Notably, external alignment with video-specific V-JEPA2 (Bardes et al., 2024) and Depth Anything 3 (Lin et al., 2025) actually *harms* performance relative to vanilla flow matching. We hypothesize that temporal relations are harder to learn than spatial ones, making objective misalignments harder to bridge. Moreover, video temporal redundancies allow models to exploit shortcuts by copying across frames rather than learning meaningful semantics, a behavior our masking mechanism naturally discourages.

*Table 2.* **Quantitative results on text-to-image generation.** Our method outperforms all external and internal alignment methods.

| Model | Steps | FID↓ | sFID↓ | IS↑ | Pre.↑ | Rec.↑ | FD-DINO↓ | CLIP↑ |
|---|---|---|---|---|---|---|---|---|
| *Without external representations* | | | | | | | | |
| Vanilla Flow | 1M | 4.08 | 8.16 | 20.49 | 0.62 | 0.64 | 204.49 | 30.66 |
| SRA | 1M | 3.70 | **8.05** | 21.00 | 0.63 | 0.64 | 176.79 | 30.78 |
| **Ours** | 1M | **3.61** | 8.14 | **21.19** | **0.64** | **0.65** | **167.98** | **30.88** |
| *With external representations* | | | | | | | | |
| REPA | 1M | 3.92 | 8.20 | 21.16 | 0.63 | **0.65** | 173.35 | 30.67 |
| SigLIP 2 | 1M | 3.97 | 8.13 | 20.65 | 0.63 | 0.64 | 196.75 | 30.68 |

*Table 3.* **Quantitative results on video generation.** Most external models harm performance.

| Model | Steps | FVD↓ | FID↓ |
|---|---|---|---|
| *Without external representations* | | | |
| Vanilla Flow | 600K | 50.95 | 9.28 |
| SRA | 600K | 49.75 | 9.02 |
| **Ours** | 600K | **47.81** | **8.92** |
| *With external representations* | | | |
| w/ DINOv2 | 600K | 49.59 | 9.39 |
| w/ Depth Anything 3 | 600K | 51.52 | 9.85 |
| w/ V-JEPA2 | 600K | 53.55 | 9.91 |

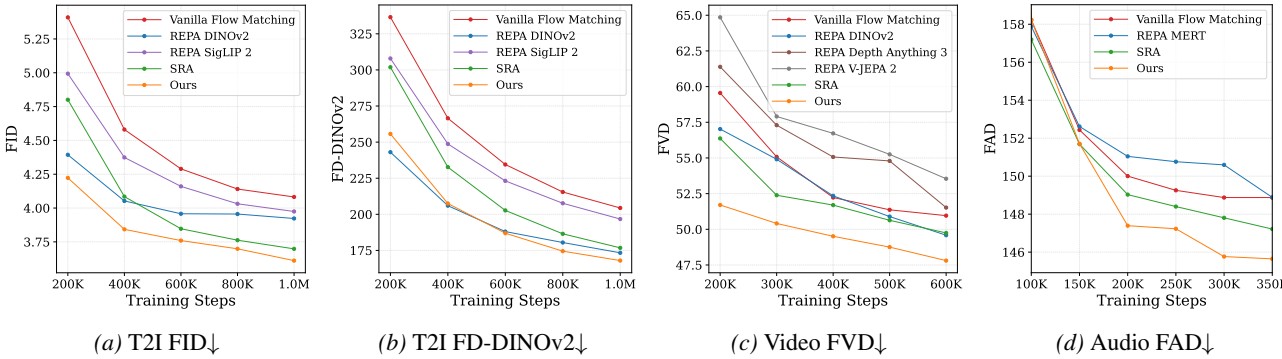

*(a)* T2I FID↓     *(b)* T2I FD-DINOv2↓     *(c)* Video FVD↓     *(d)* Audio FAD↓

*Figure 5.* **Quantitative results across modalities.** Our method significantly outperforms all external and internal alignment methods across text-based image, video, and audio generation. Our method is the only one to outperform REPA on DINOv2 FD (despite REPA directly aligning with DINOv2). Arrows indicate whether lower (↓) or higher (↑) is better.

*Table 4.* **Quantitative results on audio generation.** Our method achieves the best FAD scores across multiple CLAP variants.

| Model | Steps | CLAP↓ | CLAP-M↓ | CLAP-A↓ |
|---|---|---|---|---|
| *Without external representations* | | | | |
| Vanilla Flow | 350K | 148.874 | 0.1695 | 0.1059 |
| SRA | 350K | 147.215 | 0.1664 | 0.1034 |
| **Ours** | 350K | **145.645** | **0.1634** | **0.1001** |
| *With external representations* | | | | |
| w/ MERT | 350K | 148.883 | 0.1677 | 0.1040 |

We observe similar trends for audio (Fig. 5d, Tab. 4): our method achieves the best FAD scores across all CLAP variants, while external alignment with MERT provides no benefit over vanilla flow matching. This is further indication that external alignment struggles to generalize.

**Scaling Behavior.** We evaluate the scaling behavior of our method and REPA by training text-to-image models at four scales: 290M (deph=8), 420M (depth=14), 625M (depth=21), and 1B (depth=28). Fig. 6a shows that as we scale the model, the performance gap between our method and REPA widens consistently in our favor. Notably, our 625M parameter model outperforms the 1B REPA model, demonstrating the significant performance gains from our approach at scale. Fig. 6b demonstrates that our method exhibits consistent improvements with increased compute,

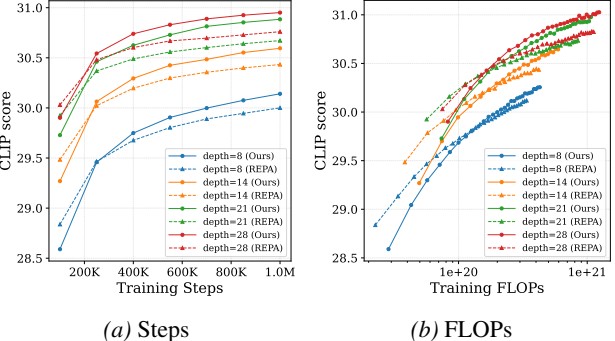

*(a)* Steps     *(b)* FLOPs

*Figure 6.* **Scaling behavior.** As model size increases (290M → 420M → 625M → 1B parameters), the performance gap between our method and REPA widens (a). Notably, our 625M variant outperforms the 1B REPA model. Our method effectively leverages increased compute, while REPA shows diminishing returns (b).

following expected scaling laws. These results validate our hypothesis that tying the model to a fixed external encoder creates a bottleneck that limits the benefits of scaling, whereas our unified framework scales as expected.

### 4.2. Multi-Modal Experiments

Having established the benefits of our approach on individual modalities, we now explore mixed and joint multi-modal setups. The former refers to a single model trained on mul-

Vanilla Flow Matching | Ours

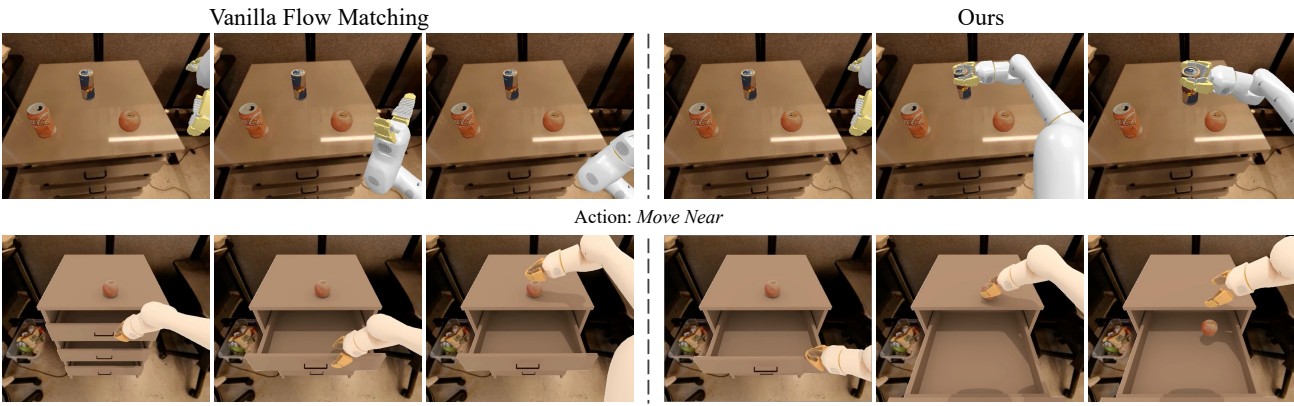

Action: *Move Near*

Action: *Place Apple in Closed Top Drawer*

*Figure 7.* **SIMPLER simulation rollouts.** Our method significantly boosts success rate for complex actions such as Move Near (top) and Open and Place (bottom).

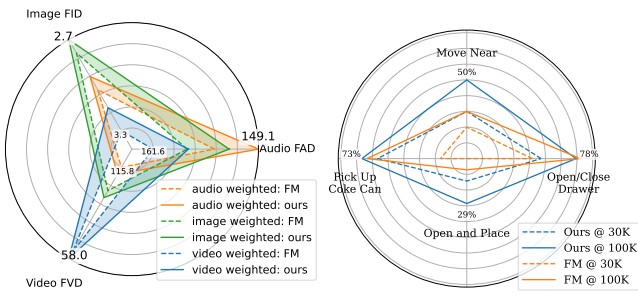

*(a)* Multi-Modal Training | *(b)* Joint Video-Action Training

*Figure 8.* **Multi-modal experiments.** (a) We train a single model on three modalities with different weightings to control the trade-off between them. Ours provides consistent improvements (shaded area) across all settings. Axes are inverted so that larger area indicates better performance. (b) Success rates for joint Video-Action prediction. Early on (30k), Ours outperforms FM across all tasks and achieves success in all task categories, whereas FM fails entirely on Open and Place tasks. Later (100k), performance on single-object tasks (Pick Coke Can, Open/Close Drawer) converges, while Ours maintains a significant advantage on complex multi-object and sequential tasks (Move Near, Open and Place).

tiple modalities, the latter to simultaneous generation of multi-modal outputs (e.g., video and corresponding audio or actions for robotic embodiments).

For the mixed-modality experiments, we follow the single-modality setup, except that we use the FLUX.2 autoencoder for T2I for enhanced visual quality. To systematically test our method's impact on each modality within this mixed setting, we employ modality-specific loss weightings $w = (w_I > 0, w_V > 0, w_A > 0)$, taking into consideration the number of samples observed for each modality (additional details in App. B). The overall loss at each step is a weighted-linear combination of each modality's loss, parameterized by $w$. A robust representation learning framework should yield improvements across all selections of $w$, demonstrating the method's ability to harmonize different representations under a single backbone. This requirement

is particularly challenging due to the different nature of the modalities—while audio representations are temporal and relatively low-dimensional, video data is high-dimensional and contains significant spatial and temporal redundancies. Fig. 8a shows results in a normalized radar chart with inverted axes, using the extreme weightings that favor each modality. This allows us to test the trade-off between different modalities. Our approach consistently improves performance across all three modalities simultaneously, even under extreme setups that favor a specific modality.

Next, we consider joint video-action prediction for embodied AI, where the model jointly predicts future video frames and robot actions from a conditioning image. We initialize from the video-weighted mixed-modality model and finetune on the RT-1 robotics dataset (Brohan et al., 2023) (73.5k episodes), evaluating on the SIMPLER simulator (Li et al., 2024a). We compare Self-Flow (Ours) and vanilla flow matching (FM) initializations, both finetuned under identical conditions. Fig. 8b reports the success rate by task group. Self-Flow consistently outperforms flow matching throughout finetuning, demonstrating more efficient learning from limited robotics data. Notably, while performance on single-object tasks (Pick, Open/Close) converges between methods, Self-Flow maintains a significant advantage on complex multi-object and sequential tasks (Move Near, Open and Place, see also Fig. 7), suggesting that our approach learns representations that improve complex visual reasoning. See App. F for details on this task and App. G for experiments on joint video-audio prediction.

### 4.3. Qualitative Results.

We provide qualitative comparisons in Figs. 9, 10, with additional results in the appendices (T2I: Figs. 23–26, 32–33; T2V: Figs. 27–31) and on our supplementary website, which contains the full video results and audio comparisons. Fig. 9 presents text rendering results from our multi-modal

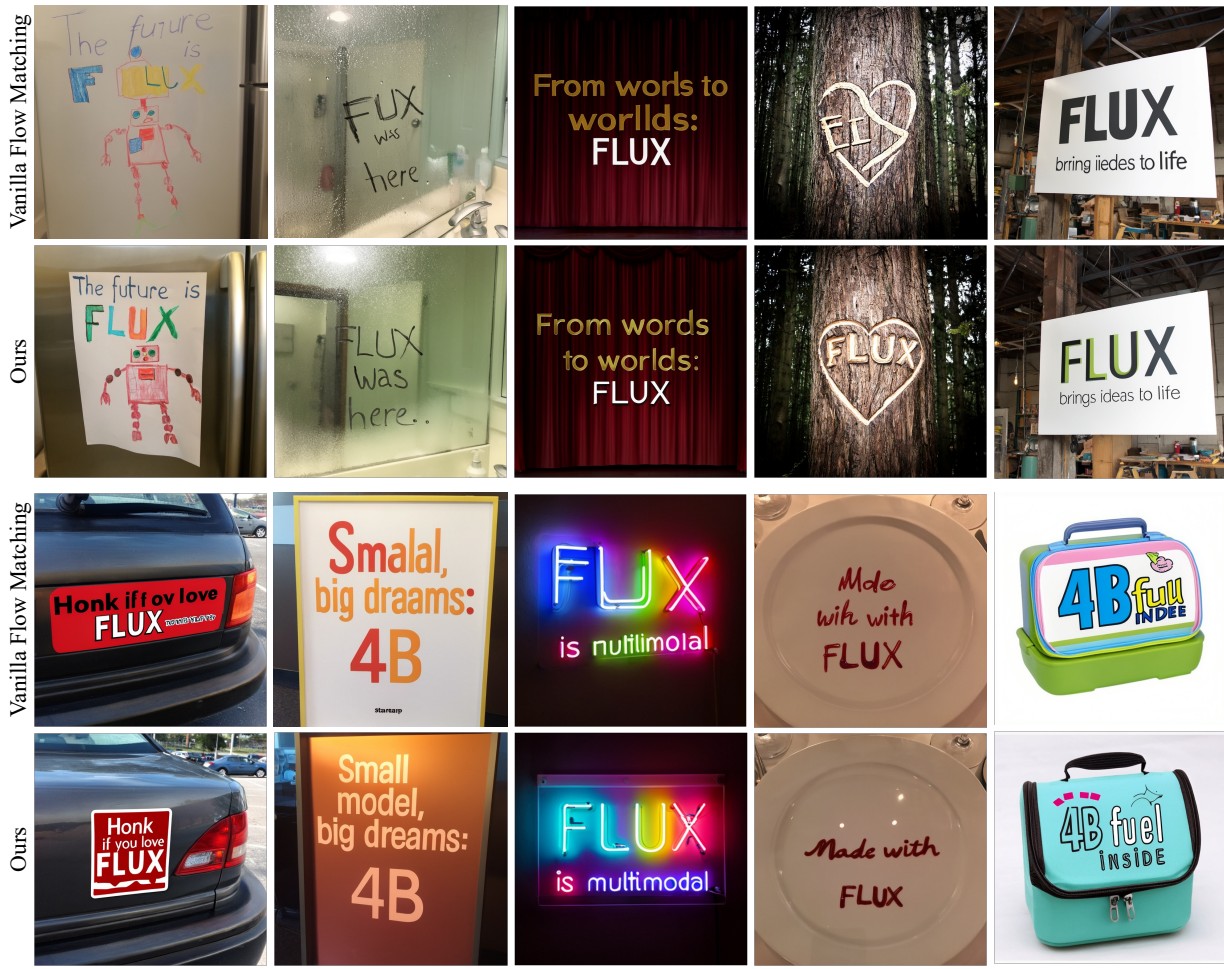

*Figure 9.* **Typography comparison.** Self-Flow significantly outperforms standard flow matching in accurate text rendering.

4B parameter model. Self-Flow significantly improves text rendering over flow matching. The comparisons in Fig. 10 show that our method consistently produces superior visual fidelity, prompt adherence, and temporal coherence over all baselines. For images, the Vespa and portrait (1st row) examples highlight improved structural accuracy and fine details compared to baselines. For video, the baselines exhibit significant structural and temporal artifacts. For example, the dancing woman (3rd row) displays limbs that spontaneously disappear. Conversely, our method maintains both spatial and temporal coherence across all samples. Notably, these video results are achieved with a modest model ($\sim$625M parameters) trained on only 6M samples, demonstrating the effectiveness of our approach in low-resource settings.

### 4.4. Ablation Study

We ablate key components of our method on the ImageNet class-to-image task (Fig. 11a). Removing the representation loss (Eq. 6) results in the most significant degradation, of over 4 points, confirming that encouraging semantic feature learning is critical for generation quality. Removing the

masking mechanism while retaining the representation loss also leads to substantial degradation of over 1 point, reinforcing the need for an explicit self-supervised paradigm beyond simple cross-layer feature alignment. Constraining the second timestep to be only slightly cleaner than the base timestep ($s \in [t, t - 0.2]$) results in degradation nearly equivalent to removing masking entirely, indicating that the formulation of masking matters: our strategy, which samples both timesteps from the full noise distribution, preserves the marginal noise distribution per token and strikes an effective balance between the generation and representation objectives. Finally, replacing the cosine similarity objective with an $\ell_1$ loss leads to numerical instabilities as training progresses due to increasing feature norms, resulting in an increase in FID at later training steps.

## 5. Limitations and Future Work

This work challenges a common assumption: that generative models require external, domain-specific encoders to improve representations and generation quality. As we show, external alignment can exhibit unexpected scaling behavior

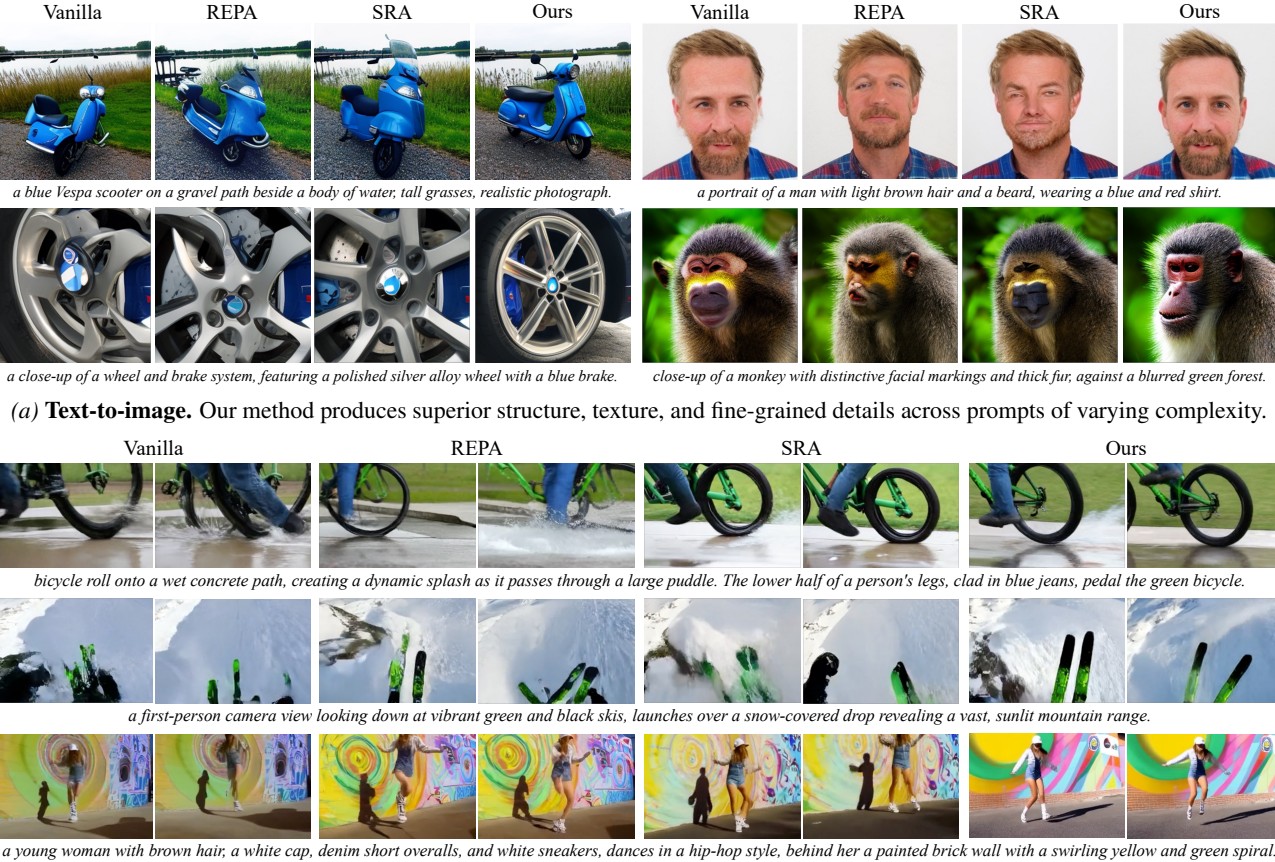

*(a)* **Text-to-image.** Our method produces superior structure, texture, and fine-grained details across prompts of varying complexity.

*(b)* **Text-to-video.** Baselines exhibit structural and temporal artifacts, while our method produces coherent and temporally smooth results.

*Figure 10.* **Qualitative comparisons** with vanilla flow matching and leading baselines: external (REPA) and internal (SRA).

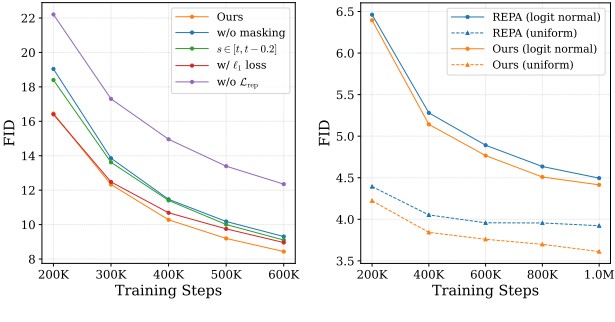

*(a)* Ablation study    *(b)* Impact of noise scheduler

*Figure 11.* **Ablations and Limitations.** (a) Removing the self-supervised loss is most detrimental. Removing Dual-Timestep Scheduling or changing the scheduling of the second timestep ($s$) significantly harms the results. (b) While better noise scheduling impacts both methods positively, our method benefits from it more.

and often struggles to generalize across modalities. For example, REPA degrades audio generation compared to vanilla flow matching (Sec. 4). Instead, we address representation deficiency at its source by unifying generation and representation learning within a single framework. This approach has trade-offs: the additional forward pass through the teacher increases training overhead, but the accelerated convergence and improved performance justify this cost (Fig. 6). Consistent with related work (Esser et al., 2024; Zheng et al., 2025), the noise scheduler $p(t)$ requires tuning as it determines masking behavior. Fig. 11b demonstrates this on text-to-image generation in the same setup described in Sec. 4.1, where a uniform scheduler outperforms a logit-normal scheduler with a shift of $\alpha = 1.78$ (App. B.2). While the better noise scheduling choice benefits both REPA and Self-Flow, the gap increases significantly in favor of the latter, which can be attributed to a more optimal timestep selection for the masking mechanism. In practice, we observe that the optimal scheduler for flow matching works well, and further tuning will likely yield additional gains.

Looking ahead, by bridging representation learning and generative modeling, our approach offers a path toward world models that harness the scalability and perceptual grounding of visual generative models without sacrificing the semantic abstraction required for planning and understanding - a direction we began to validate in Sec. 4.2. We hope this work stimulates research into consolidating generative and representation learning: two directions long pursued in isolation that may complement each other better than assumed.

**Acknowledgments** We thank the Black Forest Labs team for the codebase, architectures, and infrastructure that made this work possible. In particular, we thank Rinon Gal and Sumith Kulal for providing feedback on the manuscript and Nihanth Subramanya and Cyril Diagne for their help with the project website.

## Impact Statement

This paper presents work whose goal is to advance the field of Machine Learning. There are many potential societal consequences of our work, none of which we feel must be specifically highlighted here.

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

## A. Theoretical Discussion

We show that a pointwise minimizer of the Dual-Timestep (DTS) objective, $L_{DTS}$, is also a pointwise minimizer of the conditional flow matching (CFM) objective, $L_{CFM}$.

Let $p(t)$ denote the timestep pdf as in Sec. 3.3. The DTS forward process is:

$$
\begin{aligned}
t, s &\overset{\text{iid}}{\sim} p(t) \\
m_i &\sim \text{Bernoulli}(R_M), \quad i = 1, \ldots, N \\
\tau_i &= m_i s + (1 - m_i)t \\
x_{\tau,i} &= (1 - \tau_i)x_0 + \tau_i x_1
\end{aligned}
$$

With slight abuse of notation, let $\tau(t, s, m) \mapsto \tau$ and $x_\tau : (x_0, x_1, t, s, m) \mapsto x_\tau$ also denote the functions constructing the timestep vector forward process. Then:

$$
L_{DTS}(f) = \mathbb{E}_{x_0, x_1, t, s, m} \left\| f(x_\tau(x_0, x_1, t, s, m), \tau(t, s, m)) - (x_1 - x_0) \right\|^2
$$

We ask if a $f^\star \in \arg\min L_{DTS}$ also minimizes

$$
L_{CFM}(g) = \mathbb{E}_{x_0, x_1, t} \left\| g((1 - t)x_0 + tx_1, t) - (x_1 - x_0) \right\|^2
$$

where we generalize $L_{CFM}$ to vector timesteps by replacing $t$ with $\tau(t, 0, 0)$.

By the law of total expectation, we split $L_{DTS}$ into a homogeneous and heterogeneous component:

$$
\begin{aligned}
L_{DTS}(f) = {}& P(m = 0)\mathbb{E}_{x_0, x_1, t, s} \left[ \left\| f(x_\tau(x_0, x_1, t, s, m), \tau(t, s, m)) - (x_1 - x_0) \right\|^2 \mid m = 0 \right] + \\
& P(m \neq 0)\mathbb{E}_{x_0, x_1, t, s} \left[ \left\| f(x_\tau(x_0, x_1, t, s, m), \tau(t, s, m)) - (x_1 - x_0) \right\|^2 \mid m \neq 0 \right] =: \\
& P(m = 0)L_{hom} + P(m \neq 0)L_{het}
\end{aligned}
$$

When $m = 0$, we have $\tau_i = t$ for all $i$, so $s$ drops out:

$$
L_{hom} = \mathbb{E}_{x_0, x_1, t} \left\| f(x_\tau(x_0, x_1, t, 0, 0), \tau(t, 0, 0)) - (x_1 - x_0) \right\|^2 = L_{CFM}
$$

Since $P(m = 0) = (1 - R_M)^N > 0$, the homogeneous loss part $L_{hom} = L_{CFM}$ has a positive weight in $L_{DTS}$. Moreover, the regression target, $x_1 - x_0$, does not depend on $(t, s, m)$, so all loss contributions for a given input $(z, \tau)$ (including those from $L_{het}$) impose the same optimal value. Hence, a pointwise minimizer of $L_{DTS}$ must also minimize $L_{CFM}$, i.e. $f^\star \in \arg\min L_{CFM}$.

## B. Implementation Details

### B.1. Datasets

We use five different research datasets for our experiments. For comparisons with existing work, we use the ImageNet-1K dataset (Deng et al., 2009) comprising 1.28M training images and 50k validation images across 1k categories.

For image experiments, we use an internal research dataset of 200M images, each with four different captions of varying granularity. 50k images are reserved for validation. The text-to-image experiments use a curated subset of 20M images, while multi-modal experiments use the full dataset. We randomly choose among the available captions during training and evaluate on the most detailed captions.

Video experiments use an internal research dataset of 6M videos with a focus on motion. Each video comes with three different visual-only captions of varying granularity, one audio-only caption and one audio-visual caption. 5k videos are

reserved for validation. We randomly choose among the available captions during training, and evaluate on medium-length, visual-only captions. In the joint audio-video modeling task, we use the audio-visual captions.

For audio experiments, we use 1M audio samples with a length of 10 seconds, obtained from CC-licensed FMA data (Defferrard et al., 2017; 2018). Each sample comes with a single caption. 20k samples are reserved for validation, without overlapping tracks between training and validation splits.

## B.2. Autoencoders and Timestep Distributions

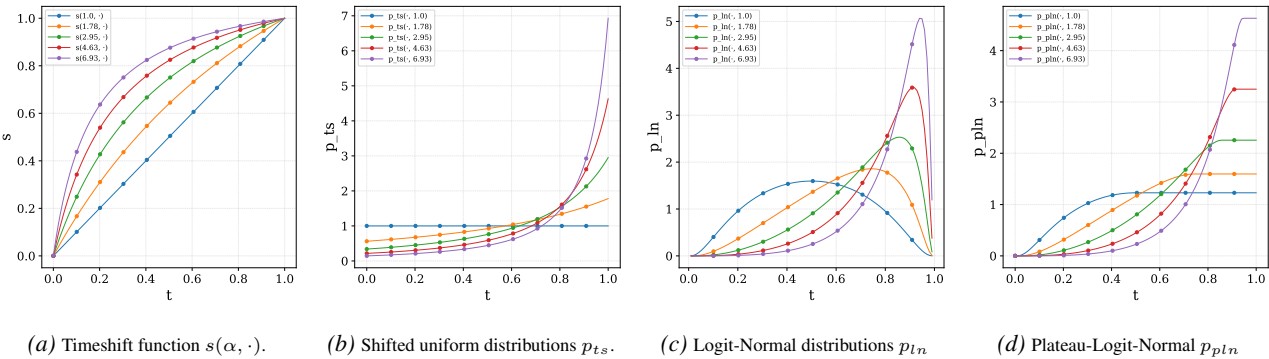

*(a)* Timeshift function $s(\alpha, \cdot)$.    *(b)* Shifted uniform distributions $p_{ts}$.    *(c)* Logit-Normal distributions $p_{ln}$    *(d)* Plateau-Logit-Normal $p_{pln}$

*Figure 12.* Shifting function and distributions used for sampling timesteps during training and shifting grids for evaluations.

Our setup generally follows the latent diffusion approach (Rombach et al., 2022b), where data is encoded through modality-specific autoencoders, and a flow-based approach (Liu et al., 2022; Albergo & Vanden-Eijnden, 2023; Lipman et al.) is used for generative modeling in this latent space. As highlighted in previous works (Kingma & Gao, 2023; Karras et al., 2022; Zeng & Yan, 2025; Hoogeboom et al., 2023; Esser et al., 2024; Zheng et al., 2025; Falck et al., 2025), the performance can be sensitive to the choice of the timestep sampling distribution used in Eq. (2).

We use the timeshift function $s(\alpha, \cdot)$ (Esser et al., 2024; Black Forest Labs, 2025) with shifting parameter $\alpha$,

$$s(\alpha, \cdot) : t \mapsto \frac{\alpha t}{1 + (\alpha - 1)t}, \tag{8}$$

to consider various choices for sampling timesteps during training. In this context, we also refer to $\alpha$ as the trainshift. Similarly, when shifting timesteps from a uniform grid with the timeshift function for evaluating models, we refer to $\alpha$ as the sampleshift.

When sampling timesteps from a uniform distribution followed by a shift with $s$, we obtain a shifted uniform distribution $p_{ts}$,

$$p_{ts}(t; \alpha) = \frac{\alpha}{(\alpha + (1 - \alpha)t)^2} \tag{9}$$

Note that the selection of evaluation timesteps also corresponds to stratified sampling from the shifted uniform distribution. When we consider the logit-normal distribution (Esser et al., 2024),

$$p_{ln}(t; \mu, \sigma) = \frac{1}{\sigma\sqrt{2\pi}} \frac{1}{t(1 - t)} \exp\left(-\frac{(\mathrm{logit}(t) - \mu)^2}{2\sigma^2}\right) \tag{10}$$

with $\mathrm{logit}(t) = \log\frac{t}{1-t}$, its samples are transformed by $s(\alpha, \cdot)$ to samples from a logit-normal distribution with shifted parameter $\mu' = \mu + \log\alpha$ (Black Forest Labs, 2025). Finally, the plateau-logit-normal distribution is obtained by keeping the probability density function of the logit-normal distribution constant after its mode. See Fig. 12 for a visualization.

For the ImageNet experiments and T2I experiments, we use the SD-VAE [1] with a uniform distribution over training timesteps to maintain comparability to previous works. To validate the applicability of our approach across different autoencoders (see

---

[1] https://huggingface.co/stabilityai/sd-vae-ft-mse

Sec. H.2), we also ran experiments on RAE (Zheng et al., 2025). While Zheng et al. (2025) used a uniform distribution with shift $\alpha = 6.93$, Black Forest Labs (2025) reported additional benefits when switching from the uniform distribution to a plateau-logit-normal distribution. In addition, we found that further increasing the shifting factor to $\alpha = 10.0$ in combination with the plateau-logit-normal distribution provides further gains.

For images in the multi-modal experiments, we use the FLUX.2 AE (Black Forest Labs, 2025) and follow their choice of a logit-normal distribution with a trainshift of $\alpha = 4.63$. For FLUX.2 AE and SD-VAE we employ a $2 \times 2$ patching resulting in a total side-length compression factor of 16 which is consistent with RAE. In this setup, images of size $256 \times 256$ are encoded to 256 tokens of dimensionality 16, 128 and 768 for SD-VAE, FLUX.2 AE and RAE, respectively.

For video experiments, we use the WAN2.2 AE (Wan et al., 2025). It uses the same spatial compression as the image autoencoders, and compresses $T$ frames into $\lfloor 1 + \frac{T-1}{4} \rfloor$ spatial latents. We train on 45 frames at a resolution of 192p, resulting in sequence lengths around 3k with dimensionality 48. To determine a suitable training timestep distribution, we train Vanilla Flow Models and search over various shift parameters $\alpha$ in combination with a logit-normal distribution. Fig. 15 shows the results from which we determined $\alpha = 2.95$ as a suitable value. In addition, in Sec. D, we explore mixing a logit-normal distribution with a uniform distribution on high noise levels, to counteract potential issues arising from the logit-normal distribution's property of vanishing density at high noise levels.

Audio experiments use the Songbloom AE from Yang et al. (2025), which mostly follows the design from Evans et al. (2025). It produces 25 latents per second of audio, resulting in 250 latents of dimensionality 64 when training on our 10 second long audio samples, or 48 latents when training on the audio track of videos. We run the text-to-audio experiments over a range of different shift values to determine good shifting values. See Sec. C.

### B.3. Architecture

When employing Dual-Timestep Scheduling, we extend the timestep conditioning of the model from a single scalar $t \in \mathbb{R}^1$ to a vector of timesteps $t \in \mathbb{R}^N$ such that each token in the sequence is conditioned on its corresponding noising timestep. To maintain comparability with previous approaches, all our ImageNet experiments utilize a SiT-XL (Ma et al., 2024) backbone comprising around 675M parameters (the exact number depending on the dimensionality of the data representation). For all other experiments, we base the design on the FLUX architecture (Labs et al., 2025), together with changes in FLUX.2[2], including the use of shared modulation layers (Chen et al., 2025a) and SwiGLU (Shazeer, 2020). We adapt the configuration to roughly match the parameter count of SiT-XL ($\sim$625M parameters). Specifically, we use a `hidden_size` of 1152, `mlp_ratio` 4, `num_heads` 16, 7 double MMBlocks and 14 single Blocks. We include bias parameters on `qkv` layers, and use 3D RoPE (Su et al., 2024) with 24 channels per dimension. For our approach and REPA variants, we follow (Yu et al., 2024) and use lightweight projection layers that add around 10M parameters. We keep track of a copy of Exponential Moving Average (EMA) (Morales-Brotons et al., 2024) weights with a decay factor of 0.9999. This copy is used as the teacher and for evaluations. We use a fixed coefficient $\gamma = 0.8$ and fixed ratios for layer selection: $l = 0.3D$, $k = 0.7D$, where $D$ denotes the depth of the model, following Yu et al. (2024); Jiang et al. (2025). Noise schedulers $p(t)$ are identical across all baselines and are selected prior to training for each task. The ratio of second timestep sampling is $\mathcal{R}_M = 0.25$ for image, $\mathcal{R}_M = 0.5$ for audio, and $\mathcal{R}_M = 0.1$ for video, due to significant temporal redundancies in video data. The qualitative samples are all obtained with 50 inference steps, using a classifier-free guidance scale of 3.5 for image generation, and 5 for video and audio generation. All quantitative metrics are reported without classifier-free guidance to ensure an unbiased comparison. The number of training steps for each modality is calibrated with respect to the data size. See Sec. I for an ablation study on these layers.

### B.4. Evaluation

On ImageNet, we evaluate SD-VAE based models using 250 SDE sampling steps as in (Ma et al., 2024). For RAE based models, we use 50 steps with a sampling timeshift of 6.93 as in (Zheng et al., 2025). In both cases, we sample 50k classes randomly, without class-balanced sampling, which (Zheng et al., 2025) reports to consistently reduce FID scores by 0.1. Scores are computed using the evaluation code[3] and reference batch from (Dhariwal & Nichol, 2021).

For other experiments, we similarly compute Fréchet distance between multivariate Gaussian distributions estimated from modality-specific representations obtained from validation data, and samples produced by a model. For video, we follow

---

[2]https://github.com/black-forest-labs/flux2
[3]https://github.com/openai/guided-diffusion/tree/main/evaluations

(Ge et al., 2024), and compute FVD (Unterthiner et al., 2018) scores using VideoMAEv2 (Wang et al., 2023) features. In addition, we include FID (framewise) scores computed using Inception (Szegedy et al., 2015) features. For audio data, we follow (Gui et al., 2024; Li et al., 2024b) and compute FAD (Kilgour et al., 2019) using the CLAP model from (Elizalde et al., 2022) (CLAP) and the music (CLAP-M) and audio (CLAP-A) variants from (Wu et al., 2024).

For non-ImageNet experiments, we sample with 50 ODE steps and a sampling shift adapted to the autoencoder. Unless stated otherwise, we use shifts $1.78$ for SD-VAE, $6.93$ for FLUX.2, $15.0$ for WAN2.2 and $6.93$ for the Songbloom AE.

### B.5. Baseline Selection

We compare against vanilla flow matching and the leading methods from both categories of representation alignment: with and without external models. For methods that use external encoders, we compare against REPA (Yu et al., 2024), the de facto feature alignment method applicable across modalities. REPA implements a generic principle: aligning intermediate representations with the hidden states of a pretrained external encoder. This allows us to plug in a domain-appropriate encoder for each of our experiments, making REPA a flexible choice across the experiments in the main paper.

Among the methods that do not employ an external encoder, we consider both SRA (Jiang et al., 2025) and Layer-Sync (Haghighi et al., 2025), which to our knowledge are the only published methods achieving representation alignment without external encoders or external supervision. We perform a preliminary experiment to select the leading baseline on a subset of 6M samples from our text-to-image research dataset (Fig. 13), where all methods share the same architecture (a 625M parameter FLUX.2 (Labs et al., 2025) backbone trained over the SD-VAE latent space), which is identical to the setup used in the main paper. We find that SRA outperforms LayerSync after 400K steps. We hypothesize that this is because LayerSync does not employ an EMA teacher or apply noise shifts to the teacher signal. Both choices weaken the distilled signal from the teacher over training and cause the trend reversal witnessed in our experiments. Therefore, we opt to use SRA for our main paper experiments. Importantly, unlike both SRA and LayerSync, our Dual-Timestep Scheduling mechanism formulates an explicit self-supervised objective directly within the flow matching framework, naturally encouraging the model to develop strong semantic representations alongside the generation capabilities, as is reflected by the results in Fig. 13 and in the main paper.

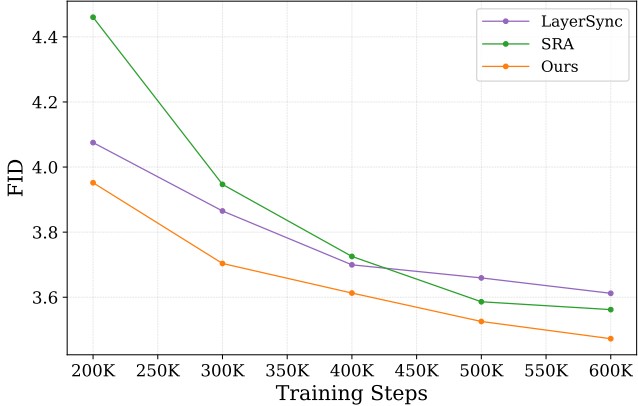

*Figure 13.* **Text-to-image results for baseline selection.** We compare SRA and LayerSync to select the leading baseline that does not employ an external encoder. We find that SRA outperforms LayerSync after 400K training steps, and attribute this to SRA's use of an EMA teacher.

## C. Additional Details on Audio Experiments

For the audio experiments in Sec. 4.1, we run all approaches with a logit-normal training timestep distribution under shifts $\alpha \in \{0.75, 1.0, 1.78\}$ and sampling shifts $\{4.62, 6.93\}$. For our approach, we include different masking ratios $\mathcal{R}_M \in \{0.05, 0.1, 0.25, 0.5\}$. To determine the best set of hyperparameters, we compute their rankings within each approach and then choose the ones with a minimal median rank across FAD (CLAP), FAD (CLAP-M) and FAD (CLAP-A). The resulting hyperparameters together with their results are shown in Tab. 5. We observe that our approach favors a slightly higher training shift of $1.0$ compared to $0.75$ favored by the other approaches. A possible explanation could be that our dual-timestep noising shifts overall SNR ratios towards the mean, thus requiring slightly higher shifts to maintain sufficient

coverage of low SNR regimes. Fig. 14 contains results from all hyperparameter runs, colored by approach. It shows that our approach not only outperforms baselines in the optimal hyperparameter setting, but instead compares favorably across a wide range of hyperparameters. Specifically, across all selections of the masking ratio $\mathcal{R}_M$, Self-Flow outperforms all baselines across all three metrics, demonstrating that it does not require careful tuning to achieve gains. With that, a judicious choice of masking ratio can boost performance further (e.g., increasing the ratio from 0.05 to 0.5 yields gains on 2 out of 3 metrics).

*Table 5.* Quantitative results on audio generation together with hyperparameters obtained as described in Sec. C.

| Model | Trainshift | Sampleshift | Masking Ratio | CLAP↓ | CLAP-M↓ | CLAP-A↓ |
|---|---|---|---|---|---|---|
| Vanilla Flow Matching | 0.75 | 6.93 | - | 148.874 | 0.1695 | 0.1059 |
| SRA | 0.75 | 6.93 | - | 147.215 | 0.1664 | 0.1034 |
| Ours | 1.0 | 6.93 | 0.5 | 145.645 | 0.1634 | 0.1001 |
| REPA MERT | 0.75 | 6.93 | - | 148.883 | 0.1677 | 0.1040 |

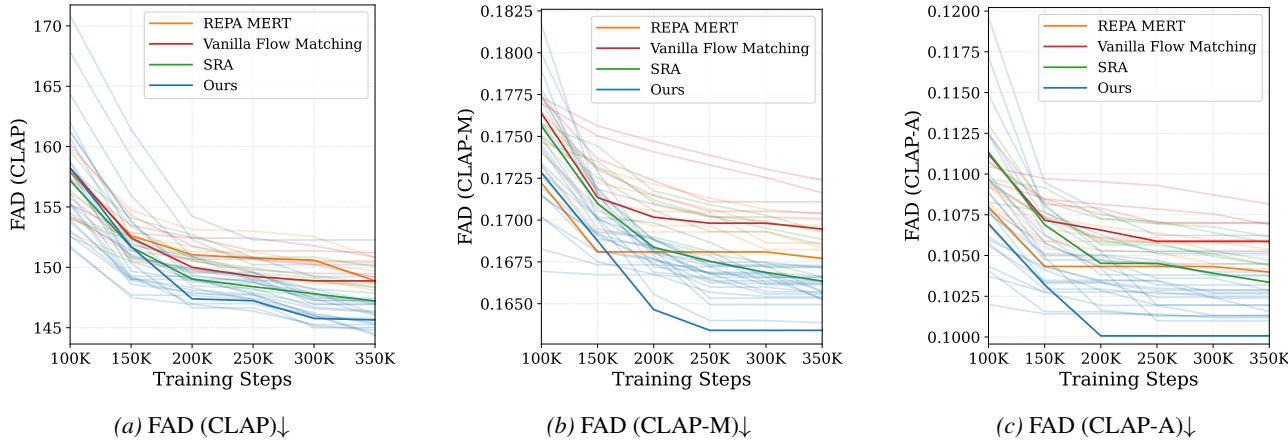

*(a)* FAD (CLAP)↓      *(b)* FAD (CLAP-M)↓      *(c)* FAD (CLAP-A)↓

*Figure 14.* Results with early-stopping from all hyperparameter variants considered in the audio experiments. The best hyperparameter runs, as determined by the selection process described in Sec. C, are shown in bold. Our approach compares favorably against all other variants across hyperparameters.

## D. Additional Details on Video Experiments

To determine the training shift parameter under a logit-normal distribution for the WAN2.2 AE, we train a Vanilla Flow Matching model with training shifts $\alpha \in \{1.0, 1.78, 2.95, 4.62, 6.93\}$. Based on the results shown in Fig. 15, we choose $\alpha = 2.95$ as the base training shift parameter on WAN2.2, which obtains the best FVD and framewise FID scores among the considered shifts.

To avoid potential issues arising from insufficient training in low SNR regimes (cf. Sec. C), we explore a low SNR modification, where for 5% of all sampled training timesteps, we sample from a uniform distribution on the high noise interval $[0.95, 1.00]$. We show results with and without this modification in Fig. 15. The performance of SRA does not change significantly under this modification, whereas we see consistent improvements for our approach and REPA DA3. REPA DINOv2 benefits in FVD but not FID and REPA V-JEPA 2's FID performance even decreases with this change. The most significant gain is observed for our FID performance, whereas our FVD performance remains best under both settings.

## E. Aditional Details on Multi-Modal Experiments

For the experiments of Sec. 4.2, we introduce minimal modifications to support training simultaneously on multiple modalities. Instead of a single set of input and output layers that project between the representation's dimensionality and the hidden dimensionality of the model, we keep one set of such layers per modality. All other weights are shared between the modalities.

In our implementation, individual mini-batches always consist of a single modality. The per-modality batch sizes are chosen such that training steps take roughly the same amount of time between different modalities. The speed is mostly determined

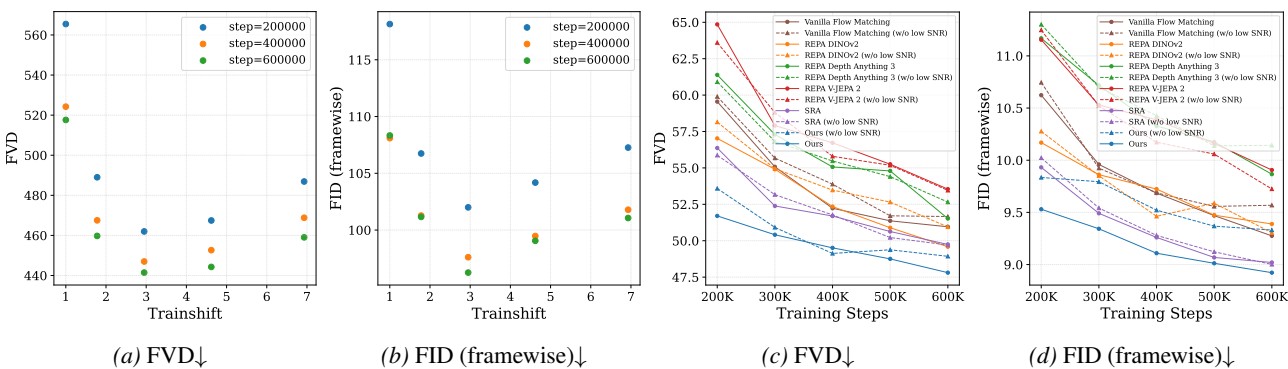

*(a) FVD↓*  *(b) FID (framewise)↓*  *(c) FVD↓*  *(d) FID (framewise)↓*

*Figure 15.* Experiments on training timestep behaviors for video training. Left two figures: Performance across different shift parameters in combination with a logit-normal distribution on a Vanilla Flow Matching model trained on WAN2.2 representations. Right two figures: Performance comparison of different approaches with and without the low SNR modification described in Sec. D.

*Table 6.* Comparison between performance of Ours and Vanilla Flow Matching (FM) in multimodal runs. For each metric, we show the values achieved by FM and Ours as well as change of Ours relative to FM. All metrics improve across all weightings.

| Weights $(w_I, w_V, w_A)$ | FID↓ | | | FID (framewise)↓ | | | FVD↓ | | | FAD (CLAP)↓ | | | FAD (CLAP-A)↓ | | | FAD (CLAP-M)↓ | | |
|---|---|---|---|---|---|---|---|---|---|---|---|---|---|---|---|---|---|---|
| | FM | Ours | Rel. Chg. | FM | Ours | Rel. Chg. | FM | Ours | Rel. Chg. | FM | Ours | Rel. Chg. | FM | Ours | Rel. Chg. | FM | Ours | Rel. Chg. |
| (0.067, 0.319, 0.614) | 4.04 | 3.69 | -8.55% | 12.2 | 11.1 | -9.59% | 72.9 | 66.3 | -9.09% | 153 | 149.8 | -2.13% | 0.115 | 0.114 | -0.47% | 0.178 | 0.177 | -0.22% |
| (0.105, 0.506, 0.389) | 3.51 | 3.25 | -7.34% | 10.9 | 10.2 | -6.60% | 64.8 | 61 | -5.85% | 155.4 | 154.2 | -0.75% | 0.12 | 0.119 | -0.74% | 0.183 | 0.182 | -0.35% |
| (0.131, 0.628, 0.241) | 3.35 | 3.19 | -4.60% | 10.3 | 9.82 | -5.04% | 62.7 | 58 | -7.40% | 160.1 | 157.4 | -1.66% | 0.127 | 0.127 | -0.57% | 0.19 | 0.189 | -0.38% |
| (0.196, 0.079, 0.725) | 3.07 | 2.99 | -2.85% | 18.1 | 17.7 | -2.01% | 116 | 112 | -3.72% | 153.9 | 149.1 | -3.08% | 0.115 | 0.112 | -3.12% | 0.178 | 0.176 | -1.44% |
| (0.308, 0.123, 0.569) | 2.83 | 2.74 | -3.06% | 15.6 | 14.8 | -5.10% | 101 | 96.7 | -4.33% | 156.4 | 152.6 | -2.40% | 0.119 | 0.117 | -1.86% | 0.182 | 0.181 | -0.74% |

by the runtime of the modality's autoencoder and the sequence length of the resulting latent representation. Specifically, we end up using batches of size 38, 8 and 16 for image, video and audio batches, respectively. Note that the requirement of approximately equal training step times is mostly a result of our implementation and could be avoided with other strategies such as sequence packing.

Since the modeling capacity has to be split among the modalities, the relative performance across modalities will be affected by the relative sampling frequencies and the relative weighting of losses computed on modality batches. Based on our dataset sizes and batch sizes, a single epoch consists of 5.26M, 0.75M and 0.0625M batches for image, video and audio, respectively. We moderately counteract this imbalance by sampling image batches with a probability of 57%, video with 30% and audio with 13%. With these ratios, a full image, video and audio epoch is reached after a total of 9.86M, 2.5M and 0.48M sampled batches.

After having fixed also the sampling ratios of modalities, the remaining lever to control which modality gets favored during training is through modality specific loss weights. We multiply the loss in Eq. (7) using different weighting factors $w_I$ for image, $w_V$ for video, and $w_A$ for audio batches. We explore various weighting triplets $(w_I, w_V, w_A)$, which can be found

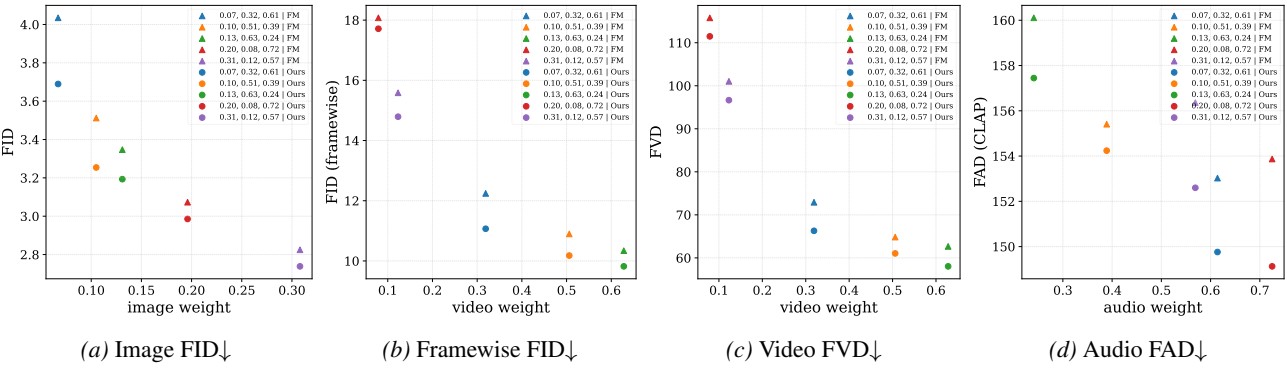

*(a) Image FID↓*  *(b) Framewise FID↓*  *(c) Video FVD↓*  *(d) Audio FAD↓*

*Figure 16.* Relationship between modality performance and weightings for multimodal runs. See Sec. E.

in Tab. 6. The *image weighted*, *video weighted* and *audio weighted* settings of Fig. 8 in the main paper correspond to the weight triplets $(0.31, 0.12, 0.57)$, $(0.13, 0.63, 0.24)$ and $(0.20, 0.08, 0.72)$, respectively.

Tab. 6 summarizes the results across modalities for each of the weight triplets and shows that our approach improves multimodal performance consistently for all weightings and simultaneously for all involved modalities. Fig. 16 visualizes the relationship between a modality's performance and weight. Finally, Fig. 17 shows the progress throughout training of all variants and demonstrates that the benefits of our approach are present throughout the complete training process.

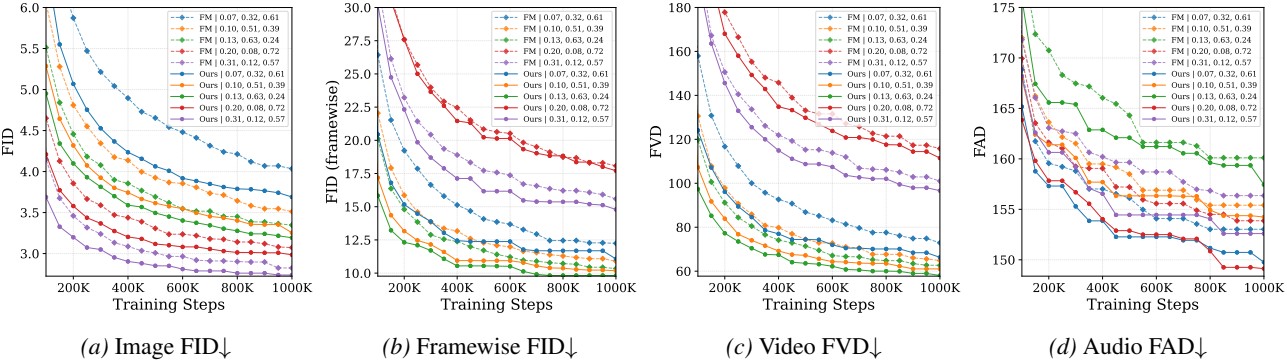

*(a)* Image FID↓    *(b)* Framewise FID↓    *(c)* Video FVD↓    *(d)* Audio FAD↓

*Figure 17.* Per-modality performance progress for multimodal runs. The benefits of our approach are present throughout the complete training process.

# F. Joint Video-Action Prediction

Recent works have demonstrated that video generation models can serve as effective action prediction models for embodied AI (Du et al., 2023; Wu et al., 2023; Black et al., 2023; Cheang et al., 2024; Hu et al., 2025; Shen et al., 2025; Pai et al., 2025; Tian et al., 2024; Li et al., 2025; Cen et al., 2025; Liang et al., 2025; Ye et al., 2026). We follow a simplified setup inspired by VideoVLA (Shen et al., 2025): instead of the full Open X-Embodiment (OXE) dataset (Collaboration et al., 2025), we only use the 73.5k episodes from RT-1 (Brohan et al., 2023), which is a small subset of OXE. Except for the 1D gripper action values, we normalize the remaining 6D action vector using 1% and 99% quantiles.

We add a new set of input and output layers to the model, which we initialize from the video-weighted runs (with 625M parameters) described in Sec. E, in either the FM (vanilla Flow Matching) or Ours variant. We encode the first frame using the image autoencoder (FLUX.2 (Black Forest Labs, 2025)) and 49 frames using the video autoencoder (WAN2.2 (Wan et al., 2025)). The audio input and output layers are dropped as they are not used in this experiment. We use a uniformly sampled timestep between 0 and 0.2 to apply noise augmentation to the image with the forward process. Handling of the text instruction for the task and the noise schedule for the video frames remain the same as in other experiments. In addition to the 49 video frames, we predict 6 action vectors (each 7-dimensional). We use the same noise timesteps for video frames and actions.

We train both variants for 100k steps and evaluate every 10k steps using the SIMPLER simulator (Li et al., 2024a). Given an instruction and rendering of the current state of the simulation, our model predicts 49 frames and 6 actions. We execute all 6 actions in the simulator, retrieve the updated rendering of the resulting state, and repeat the process. We run Pick Up Coke Can tasks 15 times, Open/Close Drawer tasks 18 times, Move Near tasks 10 times, and Open and Place tasks 7 times. We run every checkpoint twice over this list of tasks to get a rough estimate of the variance in success rates (shaded area in Fig. 18a).

Fig. 18a shows the success rate averaged over the four groups of tasks. Ours performs better than the baseline throughout the finetuning, showing that it learns more efficiently from a limited set of data. In Fig. 18b, we analyze the success rate by task group across an early step (30k) and the final step (100k). Early on, Ours outperforms the baseline in all tasks and achieves success in all task categories, whereas the baseline fails to perform Open and Place tasks at all. With continued training, we observe that the performance for simple tasks that involve a single object (picking a coke can, opening or closing a drawer) is similar for both methods, whereas for the more complex tasks that involve either two objects (placing one object near another) or a sequence of actions (open a drawer, then put an object into it), our method yields significantly better performance than the baseline. This suggests that our approach learns internal representations that improve complex visual

reasoning capabilities.

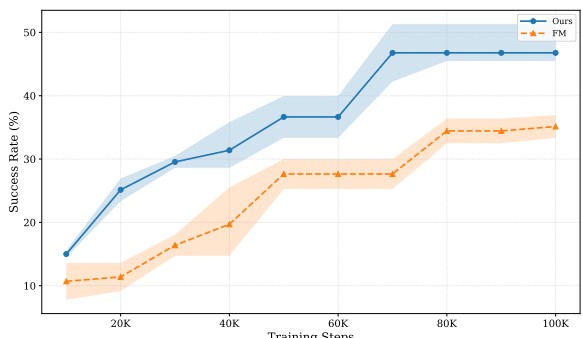

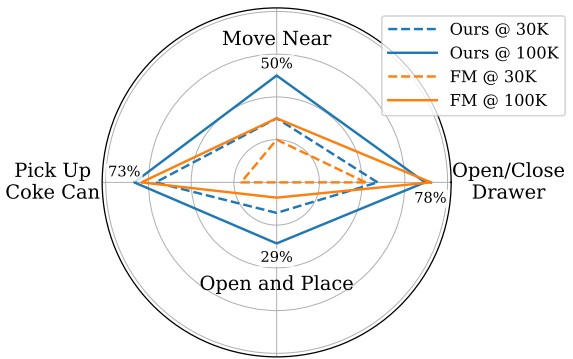

*(a)* Success rate averaged over four task groups throughout training. Ours consistently outperforms the vanilla Flow Matching baseline (FM) throughout finetuning, demonstrating more efficient learning from limited data. Shaded areas indicate variance across two evaluation runs per checkpoint.

*(b)* Success rate by task group. Early on (30k), Ours outperforms FM across all tasks and achieves success in all task categories, whereas FM fails entirely on Open and Place tasks. Later (100k), performance on single-object tasks (Pick Coke Can, Open/Close Drawer) converges, while Ours maintains a significant advantage on complex multi-object and sequential tasks (Move Near, Open and Place).

*Figure 18.* **Joint video-action prediction results on SIMPLER.** Our self-supervised flow matching framework transfers effectively to embodied AI tasks, learning more efficiently from limited robotics data and showing particular advantages on complex manipulation tasks requiring multi-step reasoning.

## G. Joint Video-Audio Prediction

We consider a joint multi-modal generation task, where we train the model for audio-video prediction from a given conditioning image. We use the first frame of each video as the condition with a noise augmentation strength sampled uniformly in $[0, 0.2]$. We consider two initializations for this task: (1) the mixed-modality model described in Sec. E with high video-weighting, and (2) a model trained only on video data.We expect the multi-modal initialization to benefit this task substantially, since it observed all modalities during training.

For each training formulation (flow matching, Self-Flow), we consider both a video-only model and a mixed multi-modal model. Note that in the case of mixed multi-modal initialization, we can reuse all the modality specific layers from Sec. E. For video-only initializations, we initialize these weights randomly instead. Fig. 19 reports the FVD of the generated videos for each initialization given the same amount of video training samples. The results confirm our intuition: mixed-modality variants consistently outperform video-only variants. Interestingly, Self-Flow with video-only initialization outperforms the baseline with multi-modal initialization, further demonstrating the strength and generality of our learned representations.

## H. Additional Image Generation Results

### H.1. ImageNet Convergence Comparison

As mentioned in Sec. 4, Self-Flow outperforms REPA on ImageNet, despite REPA using DINOv2 for feature alignment. In Fig. 20a, we present the full convergence comparison between Self-Flow and the baselines, in the same format used in (Yu et al., 2024). Since DINOv2's training set (LVD-142M) explicitly includes ImageNet and 142M additional web-retrieved images curated for ImageNet similarity (Oquab et al., 2024), it takes Self-Flow longer to catch up with REPA (compared to the other experiments in the main paper) and finally outperform it after ∼3M training steps.

### H.2. Combining Self-Supervised Flow Matching with Semantic Autoencoders

Recent works suggest integrating external representations into the latent spaces of diffusion and flow models by training the model to directly denoise the representation produced by an external encoder (Zheng et al., 2025; Wu et al., 2025a), producing semantically structured latent spaces that aim to improve generation quality and convergence speed. While these approaches are not domain-agnostic by design, we evaluate whether our self-supervised flow matching framework provides

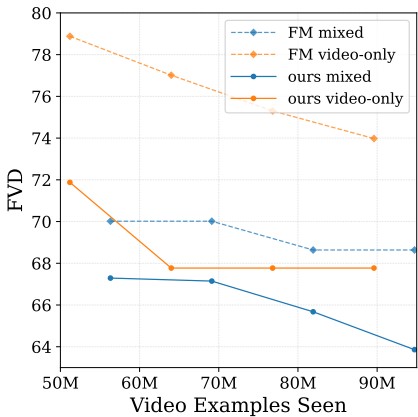

*Figure 19.* **Joint video-audio prediction** given a conditioning image, initialized from either a video-only or multi-modal (mixed) checkpoint. While multi-modal initialization benefits both formulations, even video-only Self-Flow outperforms multi-modal flow matching (FM).

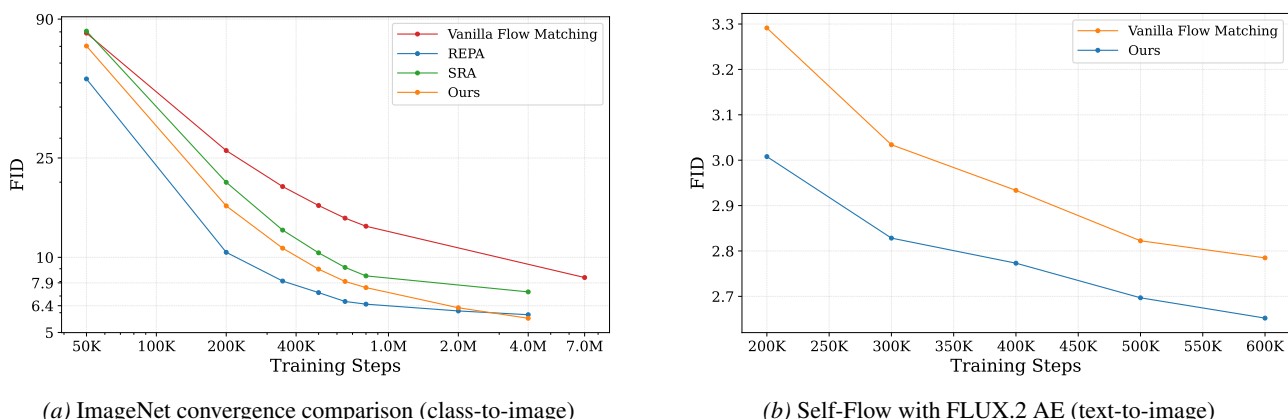

*(a)* ImageNet convergence comparison (class-to-image)  *(b)* Self-Flow with FLUX.2 AE (text-to-image)

*Figure 20.* **Additional image results. (a) ImageNet convergence comparison.** Our method outperforms REPA after ∼3M training steps, while the experiments in the main paper show an advantage to Self-Flow from the early training steps. This can be attributed to REPA's alignment with DINOv2, which was heavily trained on ImageNet. **(b) Self-Flow with semantic autoencoders.** Our self-supervised framework provides consistent improvements even when applied over semantically structured latent spaces such as RAE (Zheng et al., 2025) (on ImageNet, 4a) and FLUX.2 (Black Forest Labs, 2025) (for T2I, 20b), demonstrating complementary benefits.

additional benefits when applied over such latent spaces.

We perform experiments over RAE on ImageNet, using a plateau-logit-normal noise scheduler and a sampling shift of $\alpha = 10.0$. As shown in the main paper (Fig. 4a), our method provides consistent improvements over the RAE baseline throughout training. This demonstrates that our approach is complementary to semantic autoencoders: even when the latent space is already semantically structured, our framework further improves generation quality by encouraging the model to learn stronger representations during training. Similarly, we also observe that this observation holds in text-to-image experiments with the FLUX.2 AE (Black Forest Labs, 2025) (Fig. 20b), which uses an approach similar to REPA-E (Leng et al., 2025a) to obtain a semantically structured latent space with strong reconstruction capabilities. As discussed in Sec. 5, extending our self-supervised approach to jointly train the autoencoder in an end-to-end manner, as in REPA-E (Leng et al., 2025a), is a promising direction for future work. The consistent improvements observed across all settings in this work make it a promising candidate to yield superior results compared to methods relying on external representations.

## I. Layer Selection Ablations

We ablate the choice of student layer $l$ and teacher layer $k$ for the self-supervised objective (Eq. 6) on the ImageNet class-to-image set. In all main experiments, we use $l = 0.3D, k = 0.7D$ where $D$ denotes the total network depth.

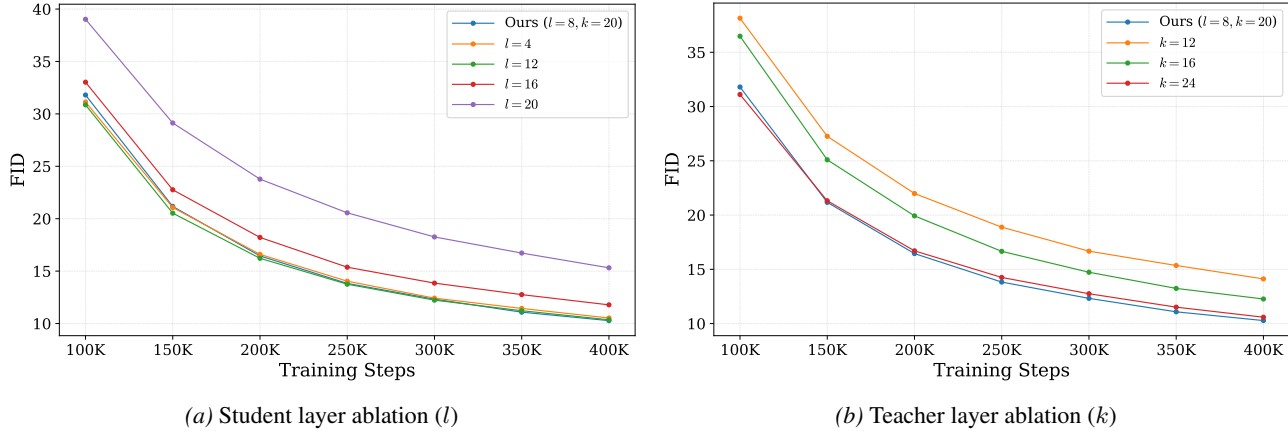

*(a)* Student layer ablation ($l$)  *(b)* Teacher layer ablation ($k$)

*Figure 21.* **Effect of layer selection.** We vary the student layer $l$ (left) and teacher layer $k$ (right) independently, with the other fixed at its default value. Performance is stable near our default selection but degrades when layers are too shallow (weak semantic signal) or when distilling to overly deep student layers (interferes with generation).

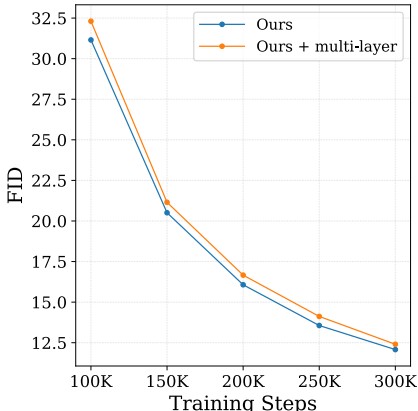

*Figure 22.* **Multi-layer alignment.** When aligning multiple student layers (here $l = 4, 8, 12$), the performance slightly degrades.

Fig. 21 shows the effect of varying each layer independently. We observe that performance is stable across a range of layer choices near our default selection (e.g., varying the student layer from 8 to 4,12 has little impact on the results), suggesting the method is not overly sensitive to this hyperparameter.

However, performance degrades when layers deviate substantially from these ranges. For the teacher, using shallow layers hurts performance because semantic representations have not yet fully emerged, reducing the effectiveness of the distillation signal. For the student, distilling to deeper layers interferes with generation quality.

We additionally explore multi-layer alignment with Self-Flow, where multiple student layers ($l = 4, 8, 12$) are aligned with the best teacher layer ($k = 20$). As can be seen in Fig. 22, this combination causes a slight degradation in Self-Flow performance. We hypothesize that aligning several layers at once encourages the model to form uniform features across layers, which harms generation.

## J. Additional Qualitative Results

We provide additional qualitative results for text-to-image generation (Figs. 23–26) and text-to-video generation (Figs. 27–31). Figs. 32–33 further demonstrate improved typography results from a scaled 4B parameter version of our multi-modal model, trained for just 100K steps on high-resolution images and videos. We refer the reader to the supplementary website for full videos, images from all figures, samples from the scaled 4B model, and audio results. For image generation, our method consistently produces improved structural coherence, texture fidelity, and preservation of high-frequency details across prompts of varying complexity. The video samples demonstrate our method's ability to produce spatially and temporally coherent results, despite being trained on only 6M video samples.

| Vanilla | REPA | SigLIP2 | SRA | Ours |
|---------|------|---------|-----|------|

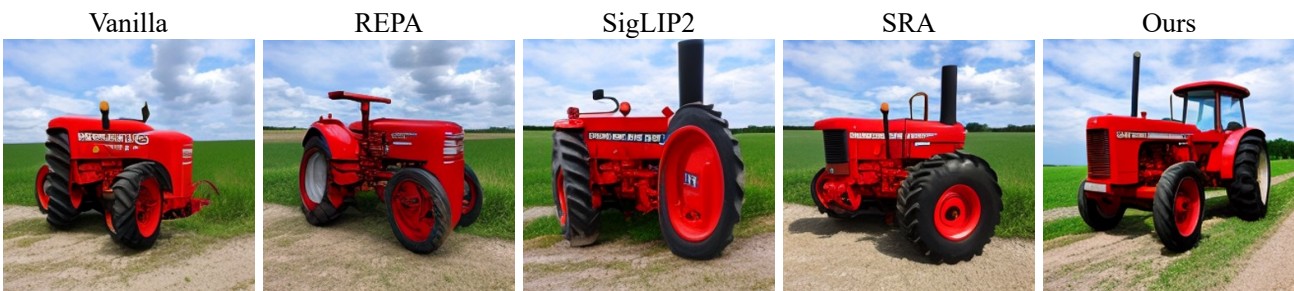

*a tractor, photographed in a vibrant red color, parked on a dirt path beside a grassy field under a partly cloudy sky, highlighting its vintage agricultural design.*

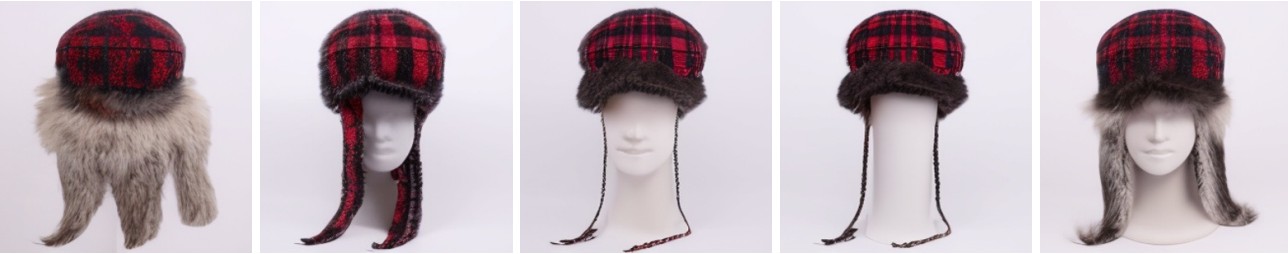

*a trapper hat, featuring a red and black plaid pattern with a thick fur around the brim and ear flaps, displayed on a mannequin head against a light background.*

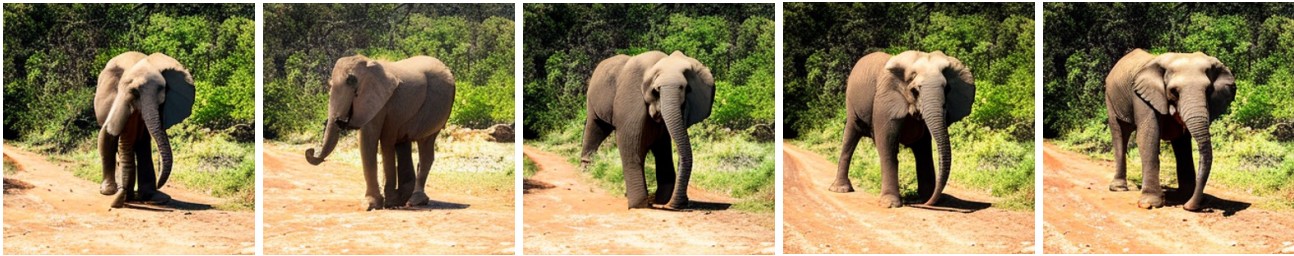

*an elephant standing on a dirt path in a natural, forested area with its trunk extended downward, captured in high-definition photography.*

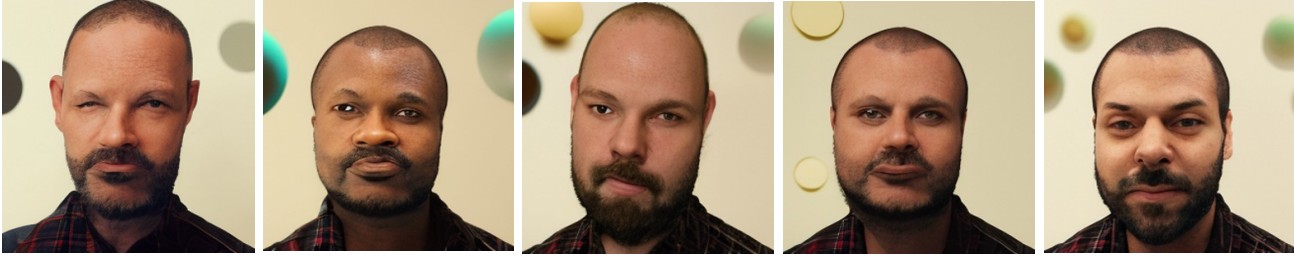

*a softly lit portrait of a man with short hair and a beard, wearing a dark shirt with a plaid collar, against a light background with blurred circular elements.*

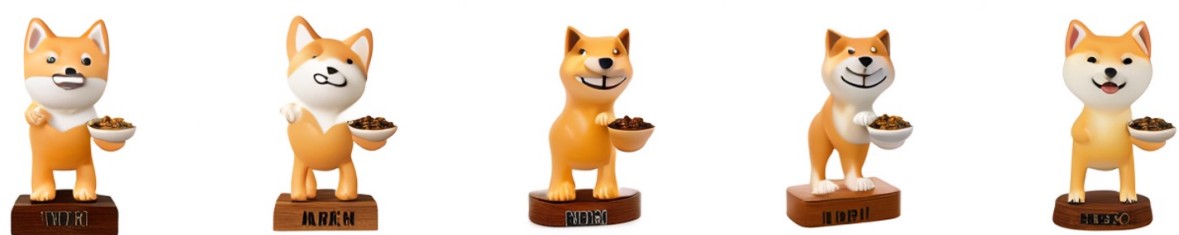

*a stylized, cartoon figurine of a Shiba Inu dog, depicted in a friendly pose while holding a bowl of food, standing on a wooden base.*

*Figure 23.* **Qualitative comparison (text-to-image).** Our method produces superior structure, texture, and fine-grained details across prompts of varying complexity.

| Vanilla | REPA | SigLIP2 | SRA | Ours |
|---|---|---|---|---|

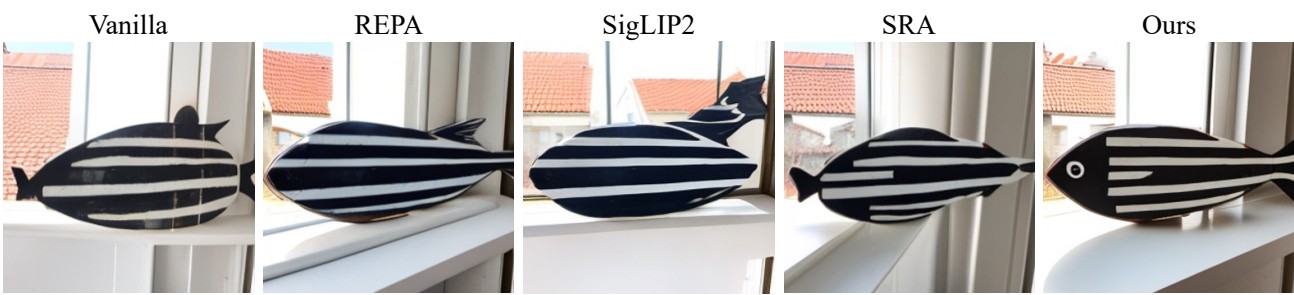

*a hand-painted, wooden fish-shaped decoration with horizontal white stripes, resting on a white shelf near a window.*

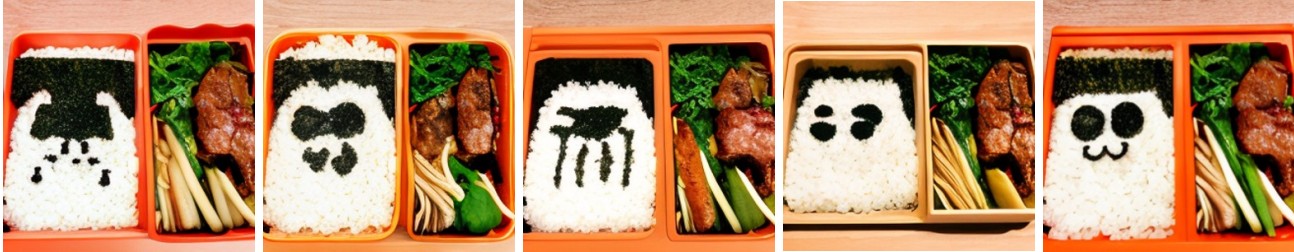

*a bento box featuring a compartment of white rice decorated with nori into a cartoonish face, by another compartment containing grilled meat and vegetables.*

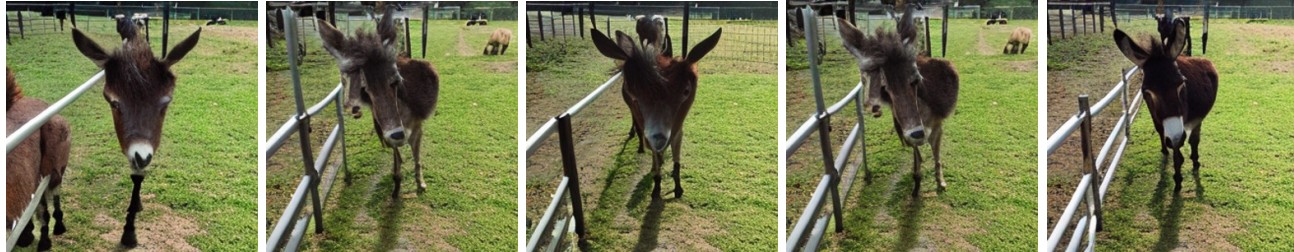

*a donkey in a grassy, fenced enclosure stands close to the metal fence. The scene is captured in natural daylight.*

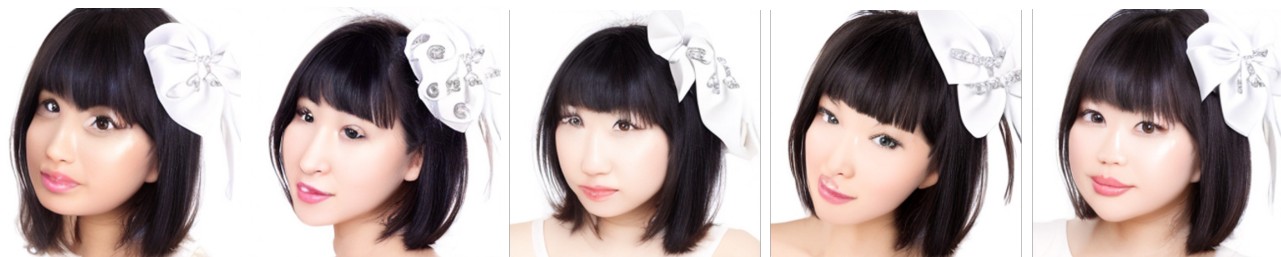

*a professionally portrait of a woman with medium-length dark hair with bangs, a large white decorative bow with silver embellishments on the side of her head.*

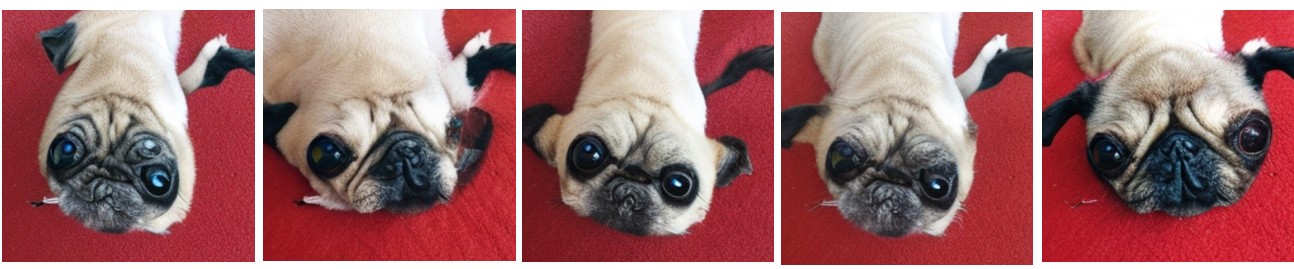

*a close-up, soft-lit photograph of a pug lying on its back on a red textured surface, with its large, open eyes looking directly at the camera.*

*Figure 24.* **Qualitative comparison (text-to-image).** Our method produces superior structure, texture, and fine-grained details across prompts of varying complexity.

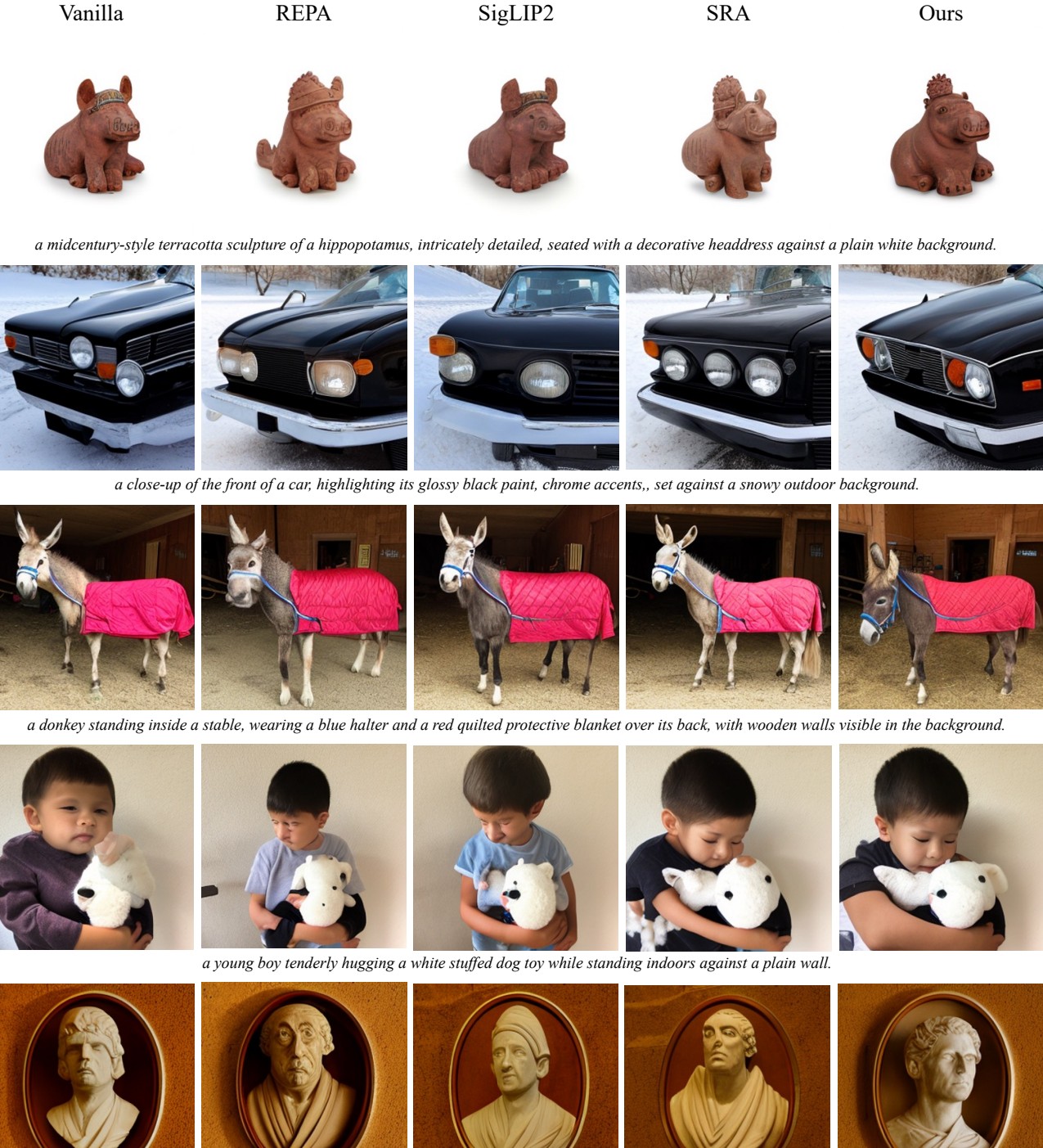

*Figure 25.* **Qualitative comparison (text-to-image).** Our method produces superior structure, texture, and fine-grained details across prompts of varying complexity.

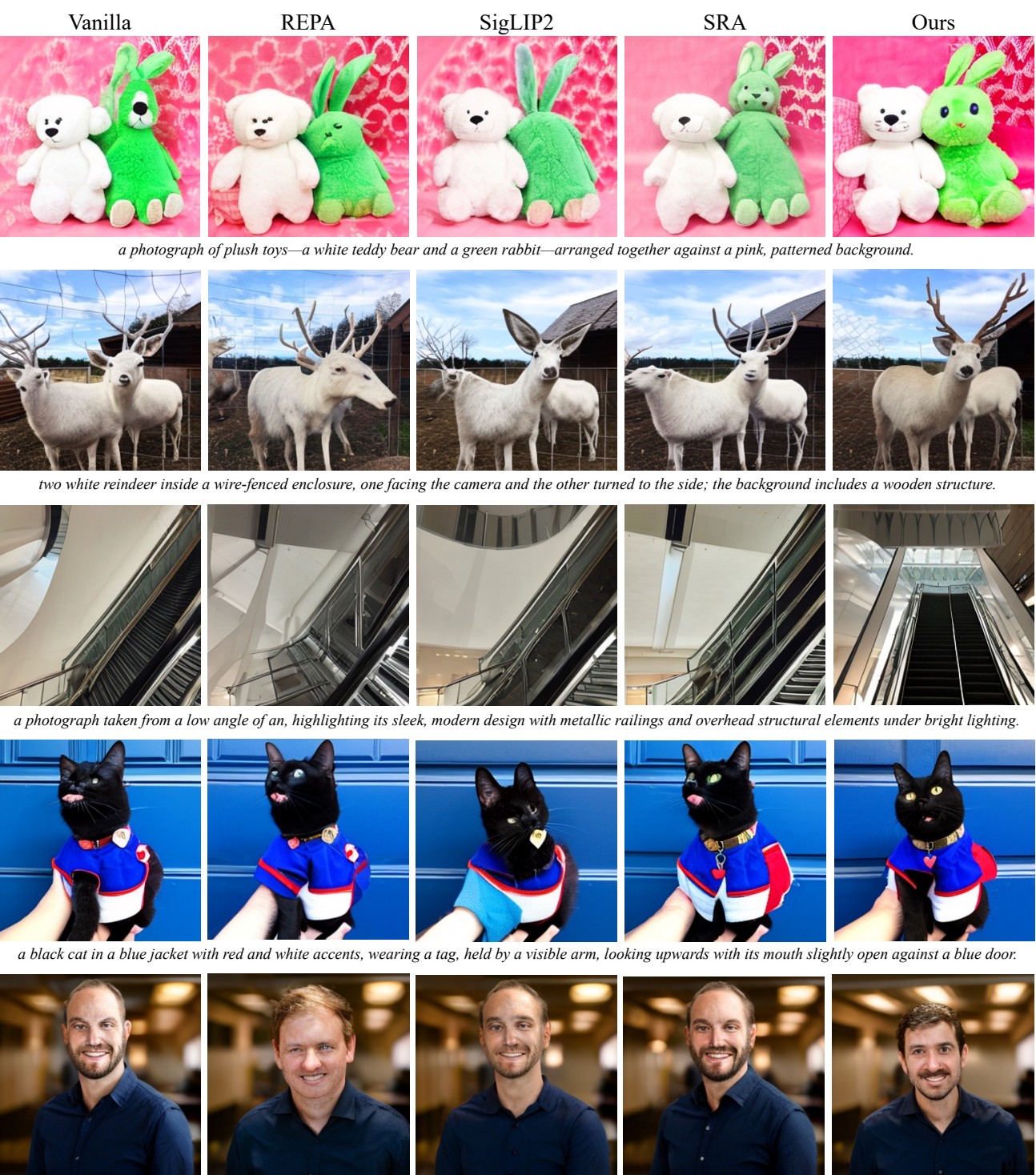

*Figure 26.* **Qualitative comparison (text-to-image).** Our method produces superior structure, texture, and fine-grained details across prompts of varying complexity.

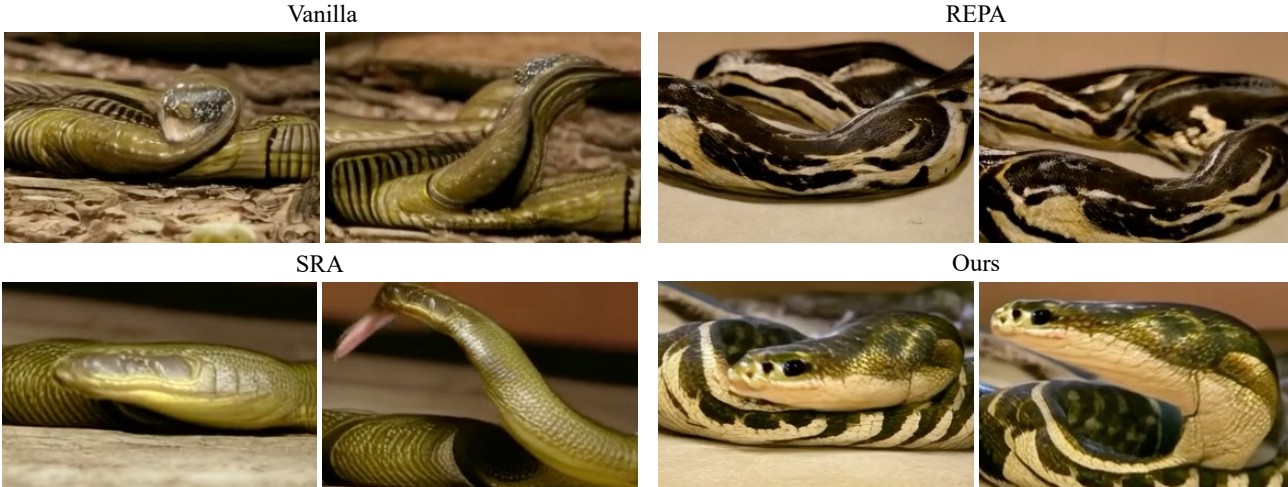

Vanilla

REPA

SRA

Ours

*a green sea turtle glides effortlessly through a colorful coral reef, filmed from a low angle. The serene creature moves with graceful flipper strokes, ancient eyes observing the vibrant marine life, shell patterns catching the filtered sunlight in the shallow tropical water.*

Vanilla

REPA

SRA

Ours

*a 3D animated orange tries to avoid being picked, hiding behind leaves, nervous expression.*

Vanilla

REPA

SRA

Ours

*a snake slowly uncoils from its resting position, massive length gradually revealed. The powerful constrictor moves with undulating grace, muscular body flowing over itself as it extends.*

*Figure 27.* **Qualitative comparison (text-to-video).** Baselines exhibit structural and temporal artifacts, while our method produces coherent and temporally smooth results.

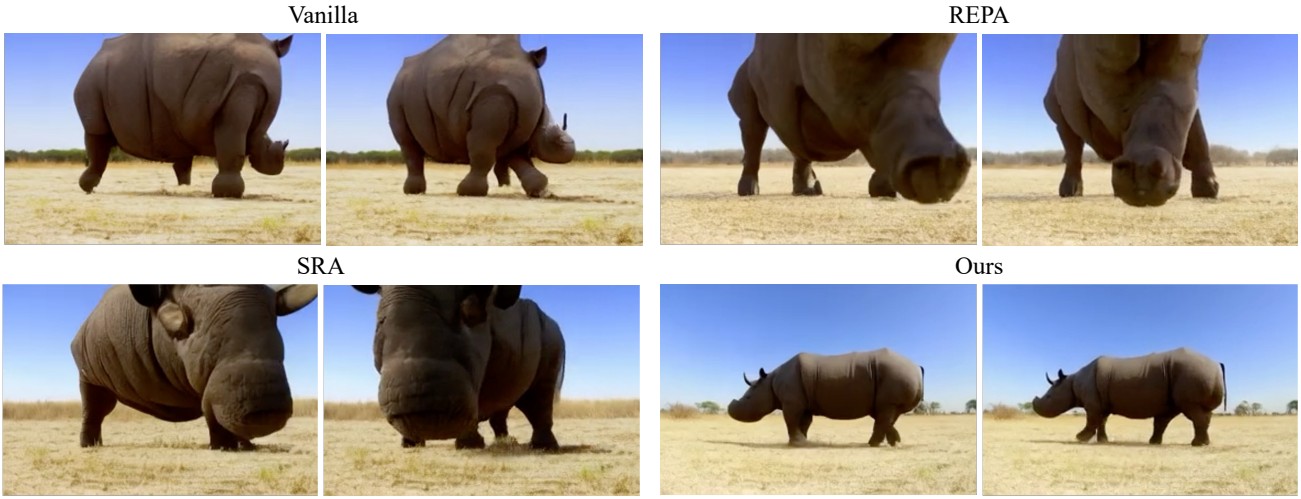

Vanilla · REPA · SRA · Ours

*a massive rhinoceros, seen from a low perspective, slowly walks across a vast, dry, light-colored plain. The powerful animal moves deliberately at a savanna landscape under a blue sky.*

Vanilla · REPA · SRA · Ours

*a cartoon 3D fox trots through fallen leaves, bushy tail swishing behind. The orange character continues on its woodland path.*

Vanilla · REPA · SRA · Ours

*a sharp chef's knife slices through a ripe red tomato on a wooden cutting board, the blade moving in smooth, confident strokes. Juice pools around, the knife work demonstrating years of practiced technique.*

*Figure 28.* **Qualitative comparison (text-to-video).** Baselines exhibit structural and temporal artifacts, while our method produces coherent and temporally smooth results.

Vanilla                                                                                    REPA

SRA                                                                                          Ours

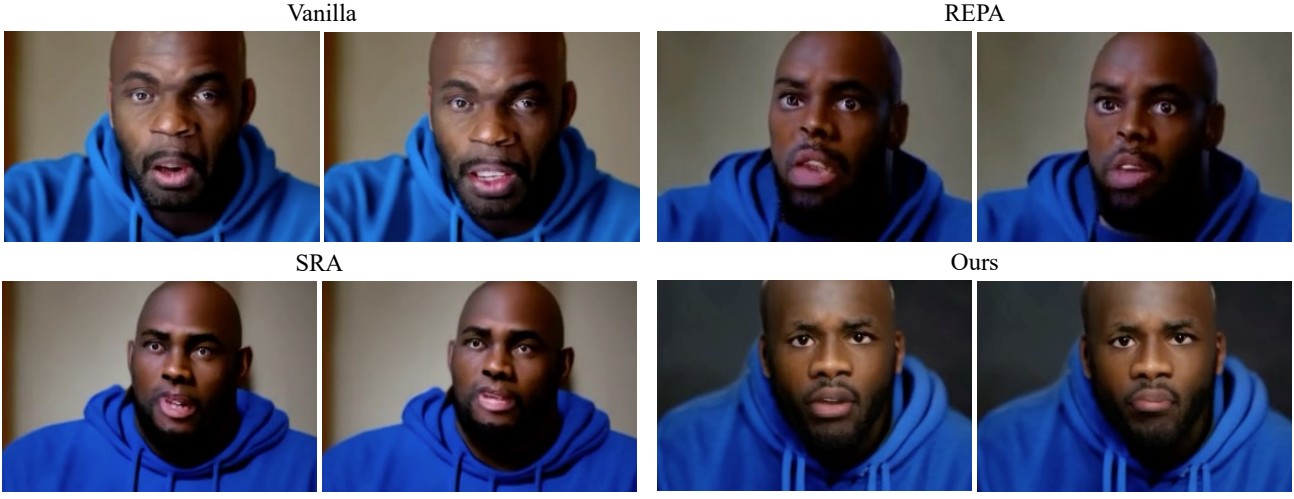

*a bald man with a trimmed beard, wearing a blue hooded sweatshirt, directs an intense and determined gaze forward, his face illuminated by soft light. His eyes are wide and mouth slightly open, conveying a focused, mid-sentence expression of direct communication.*

Vanilla                                                                                    REPA

SRA                                                                                          Ours

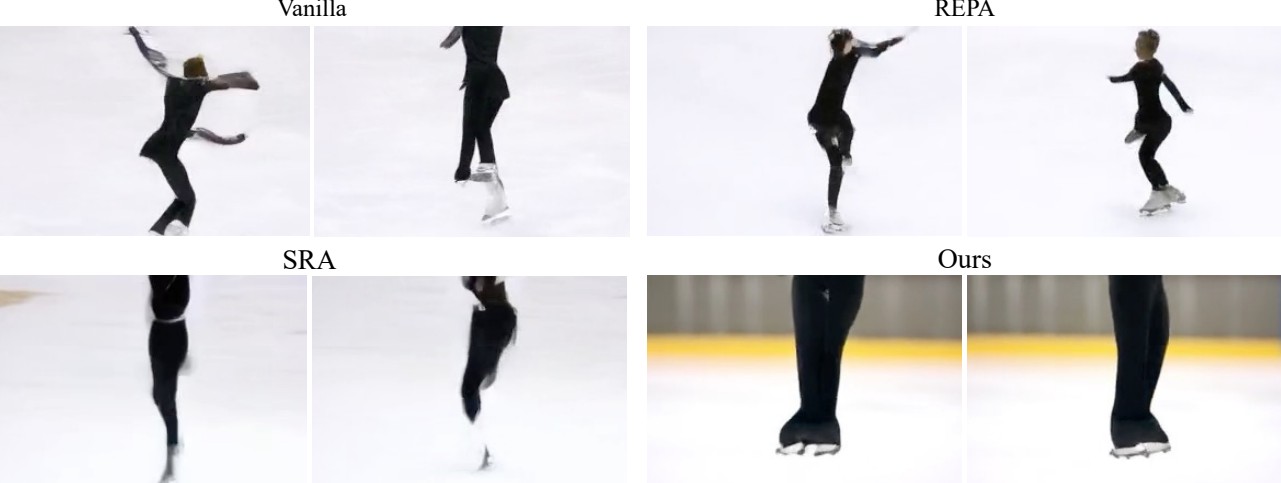

*zoom in on the legs of a figure skater completing a jump combination, blades touching down. The landing is secure, momentum carrying into the next element.*

Vanilla                                                                                    REPA

SRA                                                                                          Ours

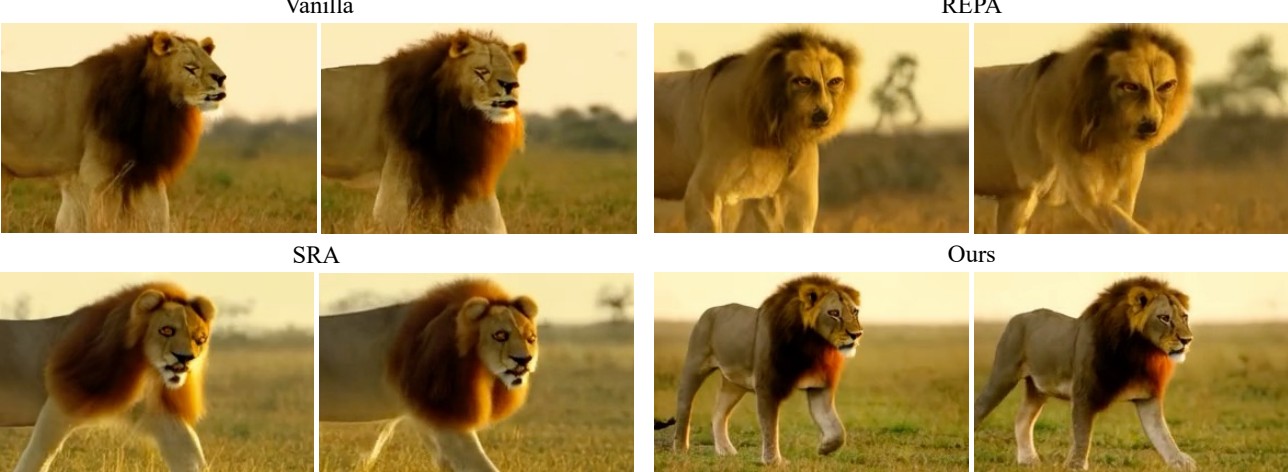

*a powerful male lion walks confidently across the savanna, captured from a low angle that conveys his regal bearing. The tawny predator moves with controlled power, mane flowing slightly in the breeze, amber eyes scanning the horizon as he patrols his territory under golden evening light.*

*Figure 29.* **Qualitative comparison (text-to-video).** Baselines exhibit structural and temporal artifacts, while our method produces coherent and temporally smooth results.

Vanilla                                    REPA

SRA                                        Ours

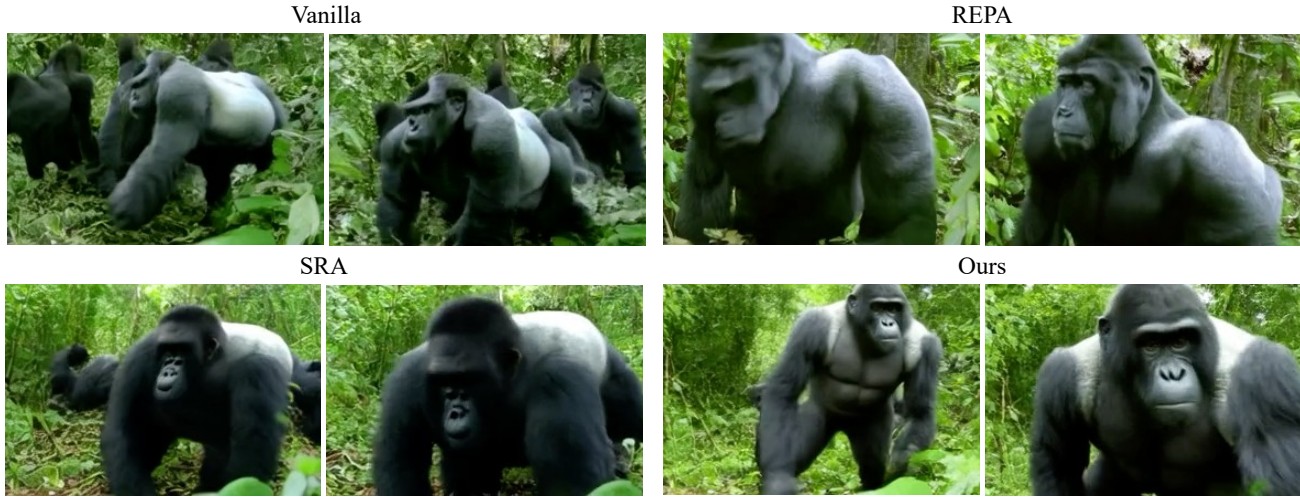

*a silverback gorilla walks through dense jungle, seen from a low, respectful distance. The massive ape moves with quiet power, silver-gray back distinctive, intelligent eyes scanning the forest.*

Vanilla                                    REPA

SRA                                        Ours

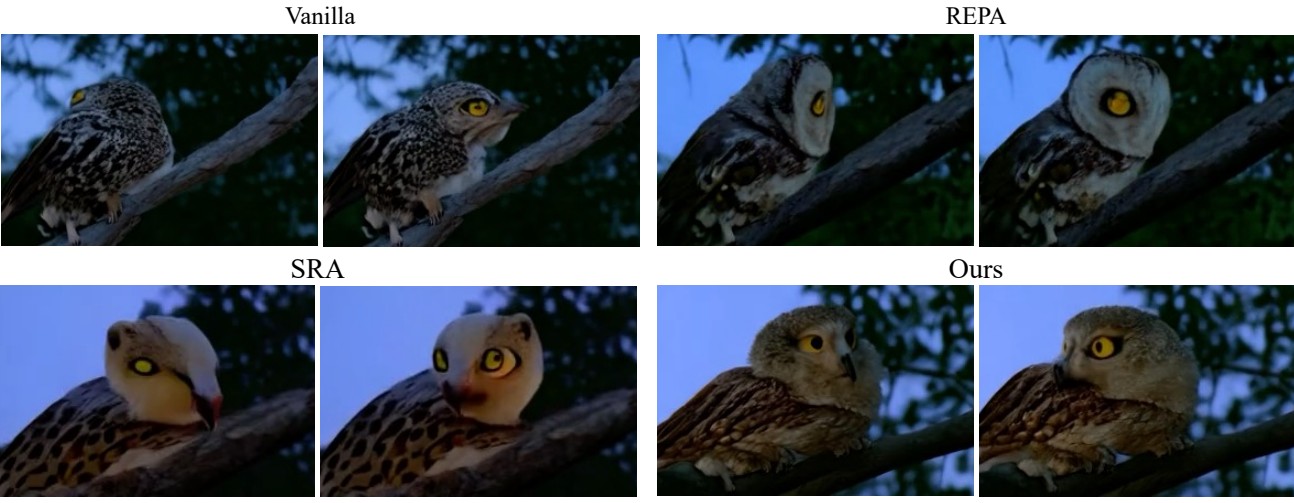

*a 3D cartoon of an owl turns its head around while perched on a tree branch at dusk, worried expression, large yellow eyes blinking slowly.*

Vanilla                                    REPA

SRA                                        Ours

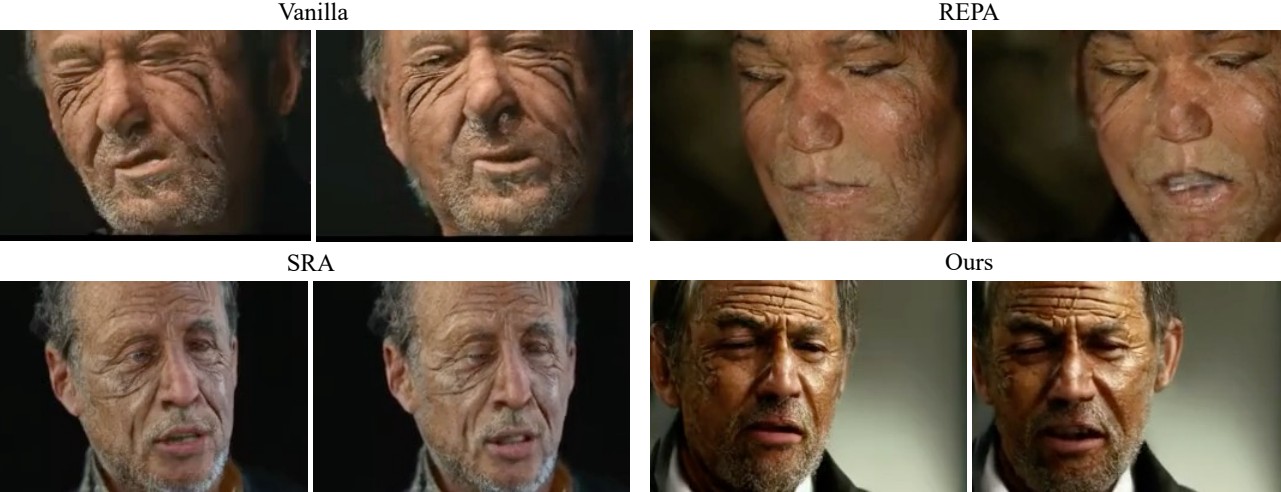

*an elderly man with deep wrinkles and a neatly trimmed grey beard speaks thoughtfully to the camera. His weathered face fills the frame as his lips form each word deliberately, occasionally pausing to gather his thoughts, warm golden lamplight casting gentle shadows across his features.*

*Figure 30.* **Qualitative comparison (text-to-video).** Baselines exhibit structural and temporal artifacts, while our method produces coherent and temporally smooth results.

Vanilla                                                    REPA

SRA                                                        Ours

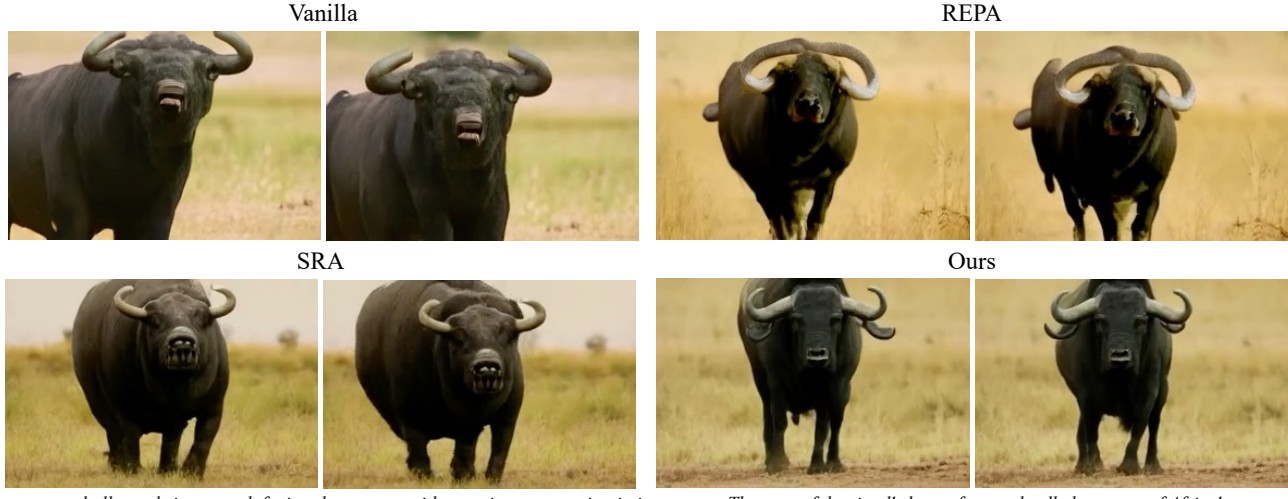

*a cape bull stands its ground, facing the camera with notorious aggression in its posture. The powerful animal's horns form a deadly boss, one of Africa's most dangerous creatures ready to charge.*

Vanilla                                                    REPA

SRA                                                        Ours

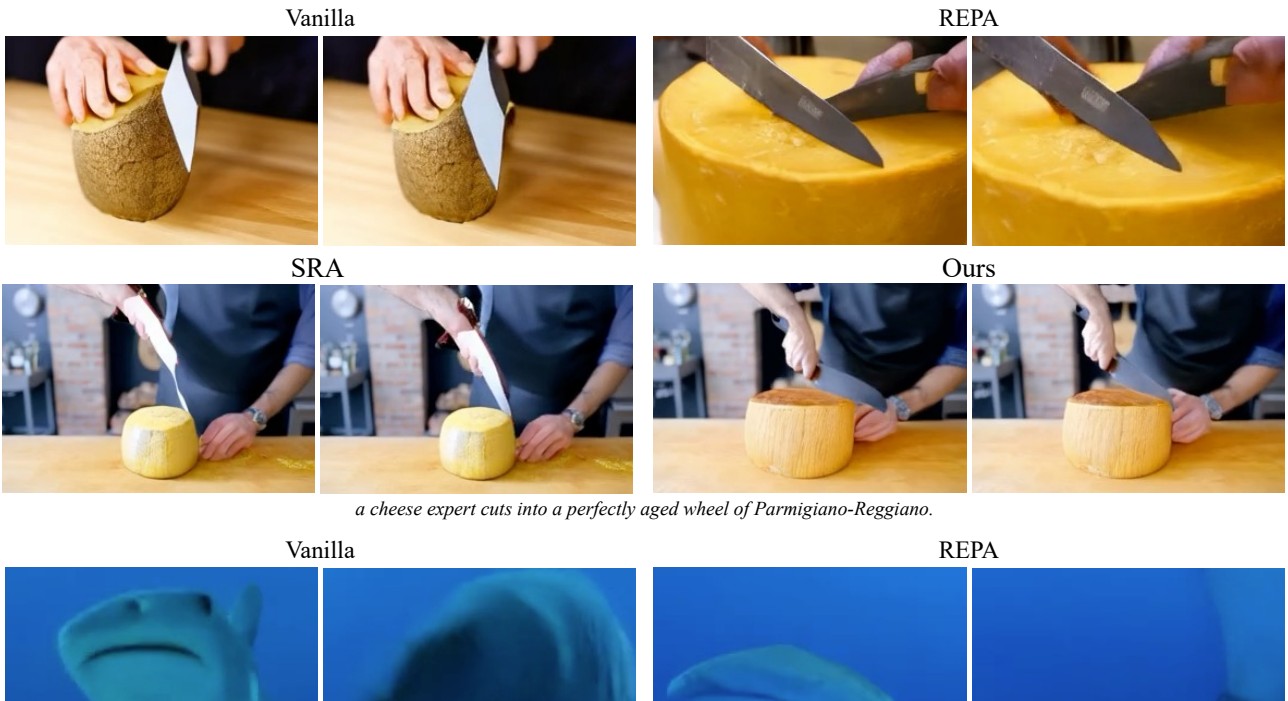

*a cheese expert cuts into a perfectly aged wheel of Parmigiano-Reggiano.*

Vanilla                                                    REPA

SRA                                                        Ours

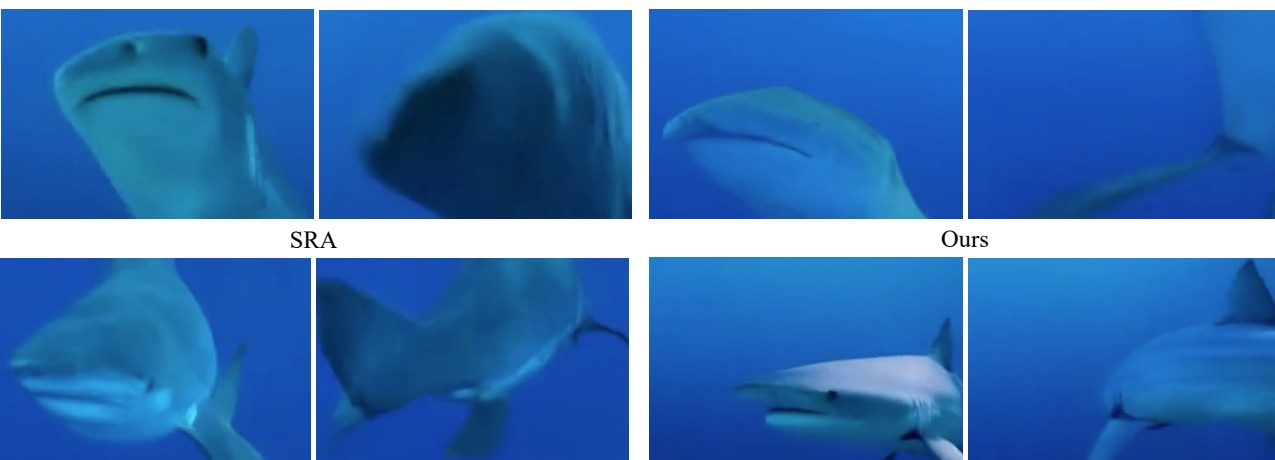

*an underwater view captures a large blue shark, starting with a medium shot of its head and front body, as it gracefully turns and begins swimming away through the deep, deep blue ocean water. The camera smoothly tracks the shark as it performs a swift turn.*

*Figure 31.* **Qualitative comparison (text-to-video).** Baselines exhibit structural and temporal artifacts, while our method produces coherent and temporally smooth results.

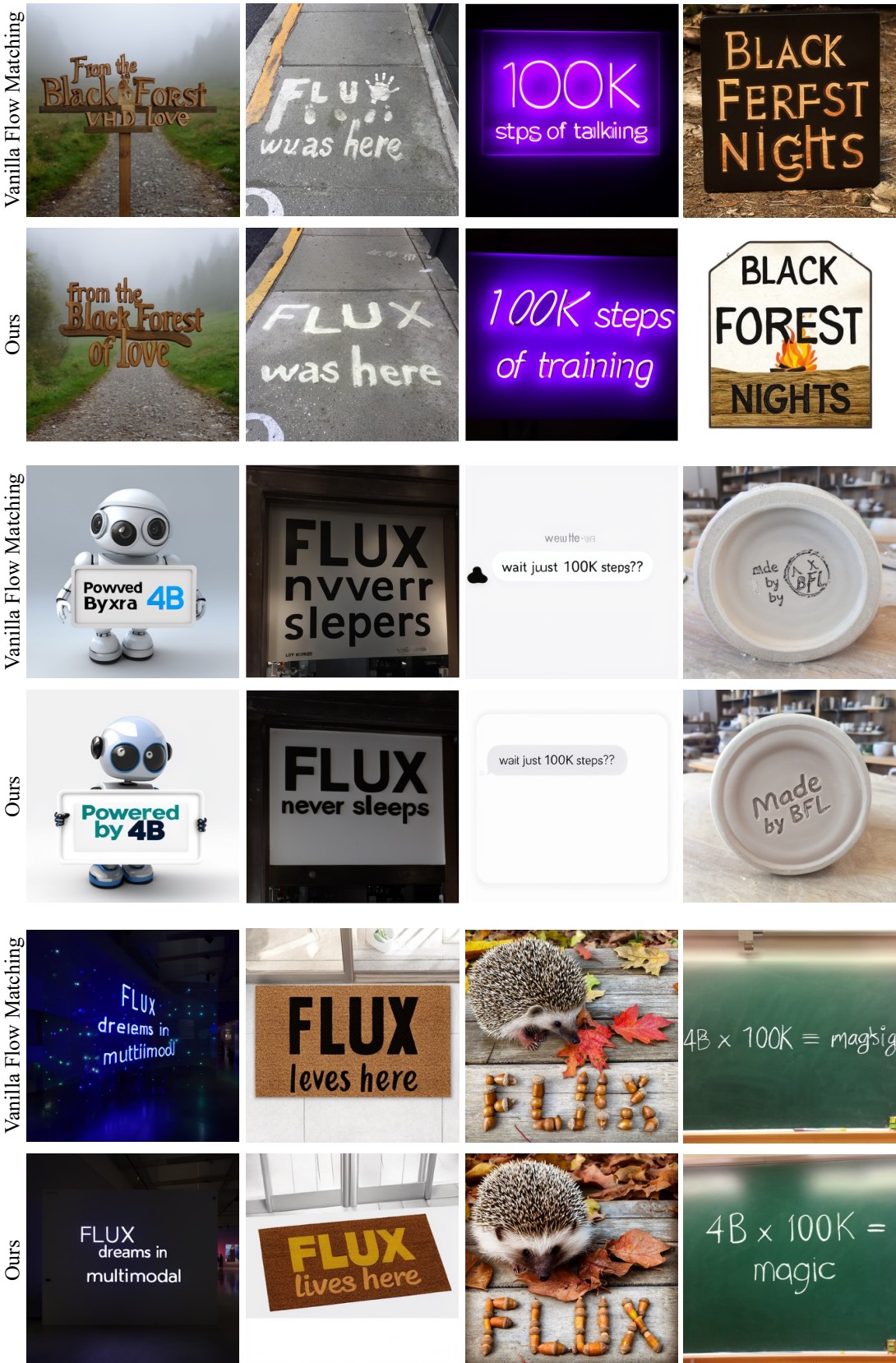

*Figure 32.* **Improved typography in text-to-image generation.** Results from our scaled 4B parameter multi-modal model after just 100K high-resolution finetuning steps. Our method produces accurate and legible text rendering compared to vanilla flow matching.

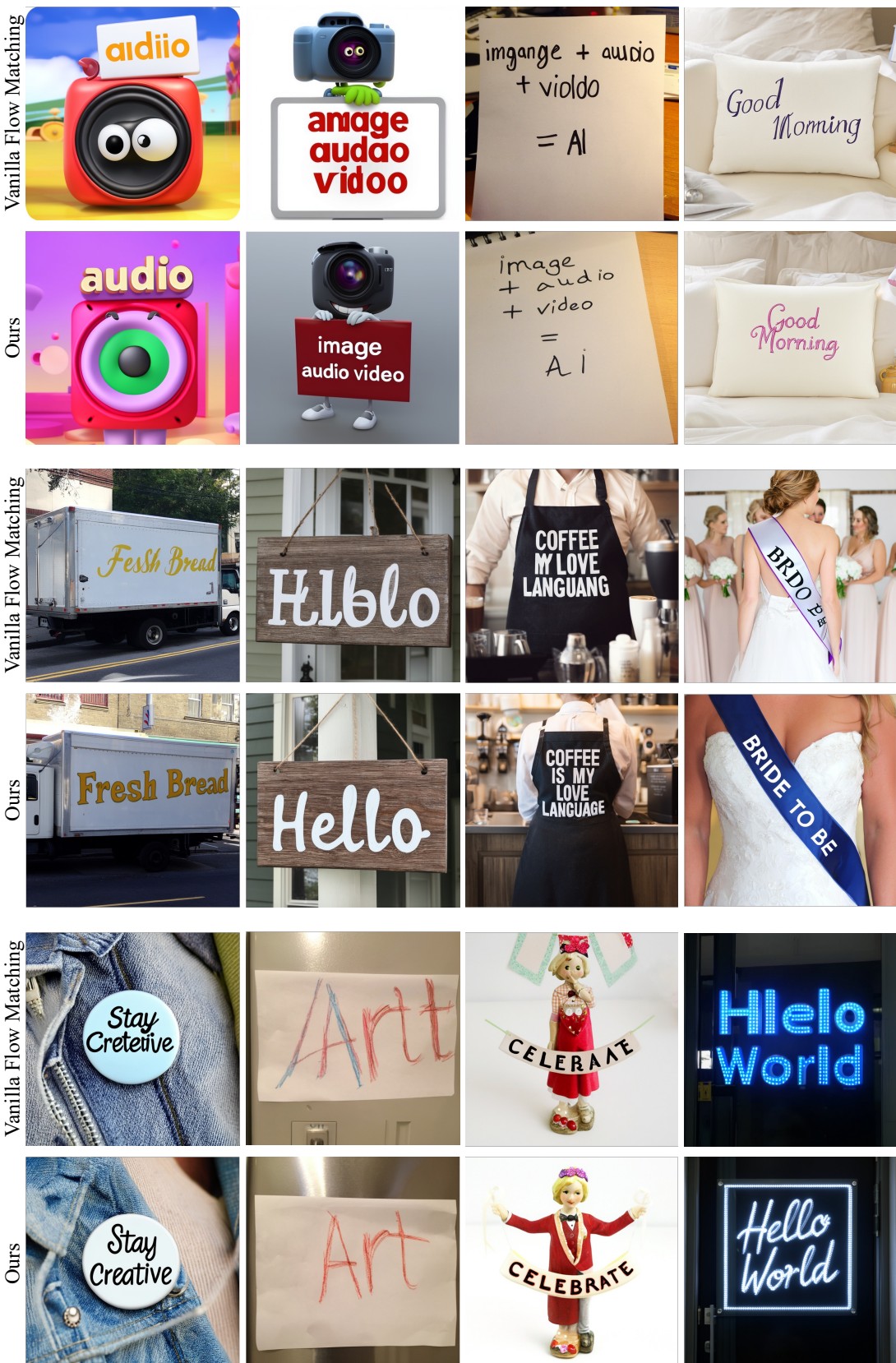

*Figure 33.* **Improved typography in text-to-image generation.** Results from our scaled 4B parameter multi-modal model after just 100K high-resolution finetuning steps. Our method produces accurate and legible text rendering compared to vanilla flow matching.

