# OpenReview forum: "Self-Supervised Flow Matching for Scalable Multi-Modal Synthesis"
_ICML.cc/2026/Conference — ICML 2026 regular_

### Official Review · Reviewer_y2Qe · 2026-02-28

**Soundness:** 3
**Presentation:** 3
**Significance:** 3
**Originality:** 3
**Overall Recommendation:** 4
**Confidence:** 3

**Summary:**

This paper proposes **Self-Flow**, a self-supervised flow matching framework that integrates representation learning directly into generative training. The authors introduce Dual-Timestep Scheduling, which applies heterogeneous noise levels across tokens to induce information asymmetry and encourage semantic inference. The method removes the need for external models and generalizes across modalities. Experiments show improved performance in image, video, and audio generation while following expected scaling behavior.

**Compliance With Llm Reviewing Policy:**

Affirmed.

**Key Questions For Authors:**

See weaknesses.

**Limitations:**

yes

**Strengths And Weaknesses:**

- **Strengths**
	- The paper is well written and easy to follow.
	- The motivation is clearly articulated, and the proposed solution is well justified and convincing.
	- The method is novel, and the experimental results convincingly demonstrate its effectiveness.
- **Weaknesses**
	- In Sec. 3.3, the connection between the two intuitive solutions and the final proposed method is not sufficiently clarified. While the proposed strategy is conceptually related to Solution #1 (masking a subset of tokens with $t=1$), its relationship to Solution #2 (assigning independent random timesteps to all tokens) is much less evident. The specific insight or motivation that Solution #2 provides for the final design is not clearly articulated.
	- It is unclear whether training the teacher model with the generative loss $\mathcal{L}_\text{gen}$​ would be a feasible or beneficial alternative.
	- Compared to approaches that rely on external models, the computational efficiency and memory usage of the proposed method are not explicitly analyzed.
	- In Sec. 3.4, the text refers to Fig. 3(b); however, Fig. 3 does not contain a subfigure labeled (b).
	- In Eq. (6), the notation $h_θ$  appears inconsistent with earlier definitions, where $f_θ$​ was introduced, suggesting a possible typographical error.

---

> ### Author Rebuttal · Authors · 2026-03-30
>
> We thank the reviewer for the positive and constructive feedback, and the interesting points for discussion. We address each question raised in the review below.
>
>
> ### __Q1: Connection between solution #2 and Self-Flow (Section 3.3):__
> The relation between Dual-Timestep Scheduling (DTS) and solution #2 (Diffusion Forcing) is that both constitute different methods of introducing information asymmetry to the flow matching paradigm. Creating such asymmetry is a key challenge in combining generation and representation learning (L. 150-160, rightmost column). There are two common approaches in the literature to achieve this: __masking a subset of tokens (masked autoencoding)__ or __assigning a different noise to different subsets of the sequence (Diffusion Forcing)__. We explore both alternatives and show empirically that they lead to severe degradation in generation quality (Fig. 2(b)). DTS is designed to avoid this failure mode, while still inducing the asymmetry needed for representation learning.
>
> ### __Q2: Training the teacher with the generative loss:__
> Please note that the teacher is an EMA version of the student (Fig. 3, L. 195-196, rightmost column). Therefore, the teacher is indeed trained with the generative loss and the representation loss combined, as it is updated using the student weights. While it is possible to consider a teacher that was trained solely using the generative objective, employing an EMA teacher is the common practice across representation learning frameworks (e.g., DINO, JEPA), as it allows the teacher and the student to share a representation space where a feature reconstruction loss is meaningful. Additionally, training an entirely separate teacher would incur significant computational costs for an additional backward pass.
>
> ### __Q3: Computational efficiency:__
> - __Compute efficiency:__ Unlike external alignment, Self-Flow scales with the model size, enabling healthy and predictable scaling laws. Fig. 7 in the paper measures compute efficiency using FLOPs, which is the standard approach to estimate computational efficiency in the literature (Peebles & Xie, 2023; Esser et al., 2024; Ma et al., 2024), and demonstrates that __Self-Flow achieves significantly better performance given the same computational budget (FLOPs)__. Following the review, we added a 1B parameter scale to Fig. 7 in the [following link](https://selfflowrebuttal.github.io/DTS_visualization/scaling.pdf). As can be observed, __the 625M parameter Self-Flow is able to outperform the 1B parameter REPA, a model with nearly twice the number of parameters__. Furthermore, the performance gap between Self-Flow and REPA widens monotonically as a function of training FLOPs across all model sizes (depth=8 through depth=28), meaning that __every additional unit of compute invested in Self-Flow yields strictly greater returns than the same compute spent on REPA__.
>
> - __Runtime analysis:__ In terms of wall clock time, let $T$ be the time for a single forward pass. A backward pass takes roughly $2T$, and the additional pass through $70$% of the teacher layers adds $0.7T$, resulting in $\sim3.7T$ vs. $3T$ for vanilla flow matching ($\sim$23% increase). The cost for external alignment is a constant independent of $T$, and its relation to the generative model’s runtime depends on the model size and external encoder selection.
> - __Memory usage:__ Self-Flow employs an EMA variant of the student as the teacher, which requires additional storage proportional to the size of the student. However, we emphasize that (i) maintaining an EMA variant is standard practice in generative training, as it typically performs better than the vanilla model. As evidence, all baselines considered in this work employ an EMA model for their evaluations, including external alignment methods. (ii) We enclose results of Self-Flow without an EMA teacher in [the following link](https://selfflowrebuttal.github.io/DTS_visualization/EMA.pdf), which demonstrate that a non-EMA variant of Self-Flow is nearly identical to the full method in later training steps, further indicating the stability of our method.
>
> ### __Q4: Figure and notation typos:__
> We thank the reviewer for pointing these out, we will make sure to fix these in the final version of the manuscript.
>
> We are happy to answer followup questions.
>
> References:
> - [1] Scalable Diffusion Models with Transformers, Peebles & Xie, 2023.
> - [2] Scaling Rectified Flow Transformers for High-Resolution Image Synthesis, Esser et al., 2024.
> - [3] SiT: Exploring Flow and Diffusion-based Generative Models with Scalable Interpolant Transformers, Ma et al., 2024.

---

> > ### Author Rebuttal · Reviewer_y2Qe · 2026-04-01
> >
> > Thank you for the authors’ response. My concerns have been fully addressed. After considering the comments from the other reviewers and the authors’ corresponding rebuttals, I will maintain my positive score.

---

> > > ### Author Response · Authors · 2026-04-01
> > >
> > > We sincerely thank the reviewer for the engagement with our rebuttal and for the continued support. We are glad the response was helpful, and we appreciate the time and care taken in evaluating our work.

---

### Official Review · Reviewer_pgjR · 2026-03-03

**Soundness:** 3
**Presentation:** 4
**Significance:** 3
**Originality:** 3
**Overall Recommendation:** 5
**Confidence:** 5

**Summary:**

This paper introduces Self-Flow, a self-supervised flow-matching framework that integrates representation learning directly into generative training via Dual-Timestep Scheduling and an EMA teacher–student objective. The method eliminates reliance on external encoders (e.g., DINO), resolves scaling pathologies observed in external alignment, and generalizes cleanly across image, video, audio, and multi-modal generation. Empirical results consistently outperform both vanilla flow matching and state-of-the-art alignment methods.

**Compliance With Llm Reviewing Policy:**

Affirmed.

**Final Justification:**

I really appreciate this paper, which introduces a self‑supervised flow‑matching framework that removes the need for external encoders (e.g., DINO), resolves the scaling pathologies associated with external alignment, and generalizes cleanly across image, video, audio, and multimodal generation. The authors have addressed all of my concerns through clear theoretical justification and comprehensive ablation studies. I strongly recommend this paper for acceptance.

**Key Questions For Authors:**

1. Does heterogeneous-noise training mismatch the uniform-noise generation trajectory?
2. How is the non-scalar time input handled with heterogeneous noise?

**Limitations:**

Yes

**Strengths And Weaknesses:**

### Strengths

- **Strong motivation and problem framing**
  The paper clearly identifies fundamental limitations of external alignment, including scaling failures and modality mismatch, and supports these claims with targeted experiments.

- **Conceptually elegant solution**
  Dual-Timestep Scheduling is simple, intuitive, and well-motivated. It avoids the train–inference gap that plagues masking-based or diffusion-forcing approaches.

- **Unified generative + representation learning**
  The method integrates self-supervision directly into flow matching without auxiliary pretraining stages or architectural changes.

- **Excellent empirical coverage**
  Results span ImageNet, text-to-image, text-to-video, text-to-audio, and multi-modal generation, with consistent gains.

- **Self-supervised method to outperform external alignment on ImageNet**
  This is a notable and nontrivial result.

- **Compatibility with semantic latent spaces (RAE)**
  Demonstrates robustness to autoencoder choice and latent representation type.

### Weaknesses and limitations

- **Limited theoretical justification**
  The paper relies heavily on empirical evidence to justify why heterogeneous token-wise timesteps do not bias the learned flow. A short theoretical discussion or controlled ablation would strengthen confidence.

- **Insufficient clarity on timestep handling**
  The treatment of token-wise timestep embeddings is implicit. Explicit clarification would improve reproducibility and reader understanding.

- **Additional hyperparameters**
  The method introduces masking ratio $R_M$ and loss weight $\gamma$, but sensitivity analysis is limited.

- **Computational overhead**
  The EMA teacher and dual forward passes increase training cost. While justified by performance gains, a cost–benefit discussion would be helpful.

---

> ### Author Rebuttal · Authors · 2026-03-30
>
> We thank the reviewer for the thorough and positive review, and for the insightful questions. We address each point below.
>
> ### __Q1: Effect of heterogeneous timesteps on learned flow:__
> **Theoretical discussion:** We show that a pointwise minimizer of the dual-timestep (DTS) objective, $L\_{DTS}$, is also a pointwise minimizer of the conditional flow matching (CFM) objective, $L\_{CFM}$.
>
> Let $p(t)$ denote the timestep pdf as in Sec. 3.3. The dual-timestep forward process is:
>
> $$
> \\begin{aligned}
> t, s &\\overset{\\text{iid}}{\\sim} p(t) \\\\
> m\_i &\\sim \\text{Bernoulli}(R\_M), \\quad i=1,\\dots,N \\\\
> \\tau\_i &= m\_i s + (1-m\_i) t \\\\
> x\_{\\tau,i} &= (1-\\tau\_i)x\_0 + \\tau\_i x\_1
> \\end{aligned}
> $$
>
> With slight abuse of notation, let $\\tau(t, s, m) \\mapsto \\tau$ and $x\_\\tau: (x\_0, x\_1, t, s, m) \\mapsto x\_\\tau$ also denote the functions constructing the timestep vector and forward process. Then:
>
> $$
> L\_{DTS}(f) = \\mathbb{E}\_{x\_0, x\_1, t, s, m} \\left\\Vert f(x\_\\tau(x\_0, x\_1, t, s, m), \\tau(t, s, m)) - (x\_1 - x\_0) \\right\\Vert^2
> $$
>
> We ask if a $f^\\star \\in \\arg\\min L\_{DTS}$ also minimizes
>
> $$
> L\_{CFM}(g) = \\mathbb{E}\_{x\_0, x\_1, t} \\left\\Vert g((1-t)x\_0 + t x\_1, t) - (x\_1 - x\_0) \\right\\Vert^2
> $$
>
> where we generalize $L\_{CFM}$ to vector timesteps by replacing $t$ with $\\tau(t, 0, 0)$.
>
> By the law of total expectation, we split $L\_{DTS}$ into a homogeneous and heterogeneous component:
>
> $$
> \\begin{aligned}
> L\_{DTS}(f) &= P(m=0) \\mathbb{E}\_{x\_0, x\_1, t, s} \\left[ \\left\\Vert f(x\_\\tau(x\_0, x\_1, t, s, m), \\tau(t, s, m)) - (x\_1 - x\_0) \\right\\Vert^2 | m=0 \\right]+ \\\\
> &\\quad\\; P(m\\neq0) \\mathbb{E}\_{x\_0, x\_1, t, s} \\left[ \\left\\Vert f(x\_\\tau(x\_0, x\_1, t, s, m), \\tau(t, s, m)) - (x\_1 - x\_0) \\right\\Vert^2 | m\\neq0 \\right] \\\\
> & \\eqqcolon P(m=0) L\_{hom} + P(m\\neq0) L\_{het} \\\\
> \\end{aligned}
> $$
>
> When $m=0$, we have $\\tau\_i=t$ for all $i$, so $s$ drops out:
>
> $$
> L\_{hom} = \\mathbb{E}\_{x\_0, x\_1, t} \\left\\Vert f(x\_\\tau(x\_0, x\_1, t, 0, 0), \\tau(t, 0, 0)) - (x\_1 - x\_0) \\right\\Vert^2 = L\_{CFM}
> $$
>
> Since $P(m=0) = (1-R\_M)^N > 0$, the homogeneous loss part $L\_{hom}=L\_{CFM}$ has a positive weight in $L\_{DTS}$. Moreover, the regression target, $x\_1 - x\_0$, does not depend on $(t, s, m)$, so all loss contributions for a given input $(z, \\tau)$ (including those from $L\_{het}$) impose the same optimal value. Hence, a pointwise minimizer of $L\_{DTS}$ must also minimize $L\_{CFM}$, i.e. $f^\\star \\in \\arg \\min L\_{CFM}$.
>
> **Controlled Ablation:** For a controlled ablation, we compare a model trained with DTS to a model trained with the vanilla conditional flow matching objective (CFM).
> Both models are evaluated on held-out data using the CFM objective under homogenous timesteps. The [results](https://selfflowrebuttal.github.io/DTS_visualization/stratified_dts.pdf) show that the model trained with DTS achieves a lower CFM loss. This indicates that DTS enables better learning of the flow. See also Fig. 2(b) of the paper showing the same comparison in terms of FID.
>
> ### __Q2: Timesteps with heterogeneous noise:__
> In standard flow matching, a scalar $t$ is broadcast to all tokens as a shared conditioning signal via adaLN modulation. With DTS, we extend this to a per-token timestep tensor $\\tau \\in \\mathbb{R}^{N}$, which introduces no new parameters. Upon publication, we will make the SiT-XL implementation, as shown [here](https://raw.githubusercontent.com/selfflowrebuttal/DTS_visualization/refs/heads/gh-pages/model.py), available.
>
> ### __Q3: Sensitivity of masking ratio and loss weight:__
> __Masking ratio:__ The submission included results across masking ratios in Supp. Fig. 12, but the sensitivity analysis was not explicitly highlighted. We now make this explicit, with results across masking ratios (0.05, 0.1, 0.25, 0.5) in [this link](https://selfflowrebuttal.github.io/DTS_visualization/masking_ratio.pdf). Across all three metrics, every masking ratio variant of our method outperforms all baselines, demonstrating that it does not require careful tuning to achieve gains. A judicious choice of masking ratio can boost performance further. E.g., increasing the ratio from 0.05 to 0.5 yields gains on 2 out of 3 metrics. The masking ratio controls the amount of clean context, and is therefore related to the redundancy of the domain, see also [this interactive demo](https://selfflowrebuttal.github.io/DTS_visualization).
>
> __Loss weight:__ All experiments use the same loss weight $\gamma=0.8$ without adjustments across modalities. This was determined by an initial sweep on ImageNet. The results are included in [this link](https://selfflowrebuttal.github.io/DTS_visualization/gamma_ablation.pdf). Increasing $\gamma$ from 0.2 to 0.5 to 0.8 is helpful, until the point where $\gamma$ overpowers the generative loss, which causes a slight degradation.
>
> ### __Q4: Computational overhead:__
> Please see the answer to Q3 of Reviewer y2Qe.

---

> > ### Author Rebuttal · Reviewer_pgjR · 2026-04-01
> >
> > The authors have fully addressed my concerns. I am satisfied with the theoretical justification that $L_{\text{DTS}}$ shares the same optimizer as $L_{\text{CFM}}$, given that the noise is added independently for each token. This insight could be incorporated into the manuscript to further strengthen the results. The ablation studies and sensitivity analyses are thorough. I decide to maintain my positive score and recommendation.

---

> > > ### Author Response · Authors · 2026-04-01
> > >
> > > We sincerely thank the reviewer for the engagement with our rebuttal and for the continued support. The helpful suggestions from the reviews, including the theoretical discussion and the sensitivity experiments, will be carefully incorporated in the final version of the manuscript.

---

### Official Review · Reviewer_gT74 · 2026-03-11

**Soundness:** 4
**Presentation:** 4
**Significance:** 4
**Originality:** 3
**Overall Recommendation:** 5
**Confidence:** 5

**Summary:**

This paper presents a self-supervised training framework that incorporates representation learning directly into flow matching models. The key idea is a dual-timestep scheduling strategy, where tokens are perturbed with different noise levels to induce information asymmetry between the student and teacher models. As a result, the student is encouraged to predict the representation of a more informative teacher token from its more heavily noised counterpart. Experiments show that this approach consistently improves performance across image, video, audio, and multimodal generation tasks, while eliminating the need for external representation encoders.

**Compliance With Llm Reviewing Policy:**

Affirmed.

**Final Justification:**

The additional clarification on the hyperparameter selection logic, the multi-layer alignment results, and the new 1B-scale results addresses my main questions well. I think the idea of inserting representation learning into generative models is quite interesting, and I look forward to seeing how this direction performs at even larger scales.

I will keep my initial positive score.

**Key Questions For Authors:**

1. Have you explored multi-layer feature alignment in Self-Flow, rather than using only one student-teacher layer pair for the representation loss? I’m curious whether aligning multiple layers improves representation learning or instead interferes with generation.

2. I am particularly interested in whether the method remains effective in a much larger, product-scale regime with sufficiently large models and long training schedules. Based on the current results, many of the baselines do not appear fully converged, so it is difficult to tell whether Self-Flow genuinely raises the  upper bound of the baseline or mainly provides faster optimization in the non-converged regime.

**Limitations:**

yes

**Strengths And Weaknesses:**

Strengths:

1. The paper is technically solid, with experiments across image, video, audio, and multimodal generation that consistently support its main claims.
2. The proposed method is simple, effective, and well motivated. In particular, eliminating the need for an external representation encoder and making the pipeline more scalable constitute significant contributions.



Weaknesses:

1. The novelty is somewhat incremental, as the proposed method is closely related in spirit to prior ideas in the self-supervised learning literature, such as self-distillation, masked prediction, and asymmetric corruption. Compared with SRA, the method also appears to use very similar hyperparameter settings(layer index), which further limits the sense of methodological distinctiveness.

---

> ### Author Rebuttal · Authors · 2026-03-30
>
> We thank the reviewer for the careful and detailed review, and for the intriguing questions. We address each point below.
>
> ### __Q1: Novelty and methodological distinctiveness:__
> Below, we address the relation between Self-Flow and two types of methods that do not employ external alignment: (i) methods that incorporate an explicit self-supervised objective (ii) methods that rely solely on the model’s learned features.
>
> __(i) Self-supervised generative methods:__ We agree that Self-Flow draws inspiration from established self-supervised principles. However, we believe that the contribution lies in the ability to integrate an explicit self-supervised framework *within* the generative framework, which is a non-trivial endeavor. As demonstrated in Section 3.3, creating information asymmetry in a way that preserves natural generative behavior is a delicate balance. Naïve approaches such as full masking or Diffusion Forcing severely degrade generation quality (Fig. 2(b)) due to train-inference mismatches (L. 185-191). Furthermore, other attempts to integrate self-supervised objectives into the generative framework (e.g., MaskDiT [1], SD-DiT [2] via masking) have fallen short of outperforming external alignment (L. 93-107, rightmost column). Self-Flow creates effective information asymmetry through Dual-Timestep Scheduling, enabling a seamless combination of representation learning and generation.
>
> __(ii) Internal feature-based methods__: First, to address the reviewer’s point on the choice of layers for alignment, we wish to clarify that this is not unique to Self-Flow or SRA, but is rather a property shared across all feature alignment methods in this family. For instance, REPA (Yu et al., 2024) uses the same student layer ($l=8$ in a depth-28 model) and demonstrates the same behavior of feature evolution across layers. This convergence in layer selection stems from the natural evolvement of semantic features in diffusion transformers: empirically, semantic understanding begins to emerge at around $30$% of the network depth, and features are most semantic around $70$% of the depth (Yu et al., 2024; Jiang et al., 2025). We therefore believe this similarity is expected and method-agnostic, and does not diminish the distinctiveness of any individual method.
> Importantly, the fact that SRA and Self-Flow share layer selections yet produce significantly different results (e.g., on ImageNet: FID of 7.27 for SRA vs. 5.72 for Self-Flow, compared to 8.3 for flow matching, Tab. 1) is evidence that the layer choice is not the source of our gains. Rather, the improvement stems from the explicit self-supervised framework introduced by Self-Flow, which goes beyond reliance on the model's naturally learned features.
>
> ### __Q2: Multi-layer alignment:__
> Preliminary results of combining multiple student layers ($l=4,8,12$) for the best teacher layer ($k=20$) on ImageNet are included in [this link](https://selfflowrebuttal.github.io/DTS_visualization/multi_layer.pdf). As can be seen, this combination causes a slight degradation in Self-Flow performance. We hypothesize that aligning several layers at once encourages the model to form uniform features across layers, which harms generation. This is consistent with Fig. 4(b), which shows that even when using $l=8$ for alignment, features continue to evolve naturally (i.e., improve) across the network up to around layer 18, suggesting that the natural progression of features across layers is important for generation quality. However, implementing such a combination is an interesting direction for future research.
>
> ### __Q3: Results at convergence/ larger scale:__
> We thank the reviewer for this question. The results reported in the paper present the performance until the baselines plateau. In the [following link](https://selfflowrebuttal.github.io/DTS_visualization/convergence.pdf) we show the effect of continued training beyond the number of steps reported in the paper compared to the most competitive baseline, REPA (for image generation, we show Fig. 1(a) from the paper, as the baseline already plateaus). As can be observed, additional training does not result in performance gains.
>
> To further illustrate Self-Flow scaling laws, we added a 1B parameter scale to Fig. 7 from the paper in the [following link](https://selfflowrebuttal.github.io/DTS_visualization/scaling.pdf). Notably, __the 625M parameter Self-Flow is able to outperform the 1B parameter REPA, a model with nearly twice the number of parameters__.
> While training production scale models using all baselines is beyond the scope of this work, we believe the results above, combined with our 4B multi-modal results (supp website), demonstrate promise and robustness at scale.
>
> We are happy to answer followup questions.
>
> References:
> - [1]  Fast training of diffusion models with masked transformers, Zheng et al., 2023
> - [2] SD-DiT: Unleashing the power of self-supervised discrimination in diffusion transformer, Zhu et al., 2024

---

> > ### Author Rebuttal · Reviewer_gT74 · 2026-04-02
> >
> > Thank you for the detailed rebuttal. The additional clarification on the hyperparameter selection logic, the multi-layer alignment results, and the new 1B-scale results addresses my main questions well. I think the idea of inserting representation learning into generative models is quite interesting, and I look forward to seeing how this direction performs at even larger scales.
> >
> > I will keep my initial positive score.

---

> > > ### Author Response · Authors · 2026-04-02
> > >
> > > We sincerely thank the reviewer for the engagement with our rebuttal and for the continued support. The helpful suggestions from the reviews, including the discussion on hyperparameter selection, the multi-layer alignment results, and the 1B-scale experiment, will be carefully incorporated in the final version of the manuscript.

---

### Official Review · Reviewer_uWbc · 2026-03-12

**Soundness:** 2
**Presentation:** 3
**Significance:** 3
**Originality:** 3
**Overall Recommendation:** 3
**Confidence:** 4

**Summary:**

The paper proposes a self-supervised flow matching paradigm to eliminate the reliance on external representation models for training generative models. The authors introduce dual-timestep scheduling, which applies heterogeneous noise levels to input tokens, creating an information asymmetry. A student network is trained to denoise the input while simultaneously matching the features of an EMA teacher network that processes a less noisy version of the same input. Experiments show the superior scalability and multi-modal generalization of the proposed method.

**Compliance With Llm Reviewing Policy:**

Affirmed.

**Final Justification:**

Thanks for your response. After carefully reading the comments, I will keep my original score.

**Key Questions For Authors:**

Please see the Weaknesses.

**Strengths And Weaknesses:**

Strengths:

+This paper study a highly relevant and timely problem of the scaling bottlenecks and cross-modality limitations of external representation alignment.

+Integrating a masked-autoencoder-style self-distillation objective natively into the continuous-time flow matching framework via Dual-Timestep Scheduling is an elegant and structurally novel approach.

Weaknesses:

-The key claim of the paper is that self-supervised alignment (Self-Flow) outperforms external alignment (REPA). However, the comparison is fundamentally confounded. Self-Flow introduces two components: DTS and an EMA Self-Distillation loss. REPA uses Standard Scheduling and External Distillation. Figure 2b explicitly demonstrate that DTS alone improves vanilla flow matching. Therefore, more detailed ablation should be conducted.

-The mask is sampled randomly token-by-token. In continuous modalities like images and videos, local spatio-temporal redundancy is exceptionally high. A heavily noised token is highly likely to be adjacent to a "clean" token. Consequently, the student network can minimize the representation loss trivially by performing local spatial/temporal interpolation, bypassing the need to learn the "global semantic connections" you claim it learns.

-The transition to multi-modal joint training relies on manually tuned, static modality-specific loss weights. For a framework claiming to be a unified, scalable paradigm, relying on brittle hyperparameter tuning to balance modalities undermines the "elegance" of the approach.

---

> ### Author Rebuttal · Authors · 2026-03-30
>
> We thank the reviewer for the feedback and the interesting points for discussion. We address the questions raised in the review below.
>
> ### __Q1: Ablations:__
> Please note that the ablations requested in the review are included in Fig. 10(a) (albeit with slightly different terminology). For example, w/o $\mathcal{L}_{{rep}}$ denotes an ablation where we keep only DTS, and remove the representation loss, w/o masking refers to the mirror ablation where we keep the representation loss but remove DTS. We enclose the same ablation study presented in the paper with only the ablations referred to in the review in the [following link](https://selfflowrebuttal.github.io/DTS_visualization/ablation.pdf) for the reviewer’s convenience. We additionally include vanilla flow matching for compatibility with Fig. 2(b) referenced in the review. As can be observed, both components have a noticeable impact on the performance of our method, and the full benefit of it can only be obtained by combining the two. This relation is expected, since __DTS serves as the basis to create the asymmetry required for effective representation learning__. Thus, the two are fundamentally tied to each other.
>
> While the use of an EMA teacher is the common practice in self-supervised methods, we additionally enclose an ablation comparing the performance of Self-Flow with and without it in the [following link](https://selfflowrebuttal.github.io/DTS_visualization/EMA.pdf). As can be expected, the non-EMA variant begins with a lead since EMA catches up gradually with the student in earlier training. As EMA catches up, it obtains a slight advantage over non-EMA. Notably, this advantage is not very significant, indicating that our method’s advantages stem from DTS and the self-supervision more than the use of an EMA checkpoint.
>
>
> ### __Q2: Masking effectiveness in continuous modalities:__
> The need to learn the "global semantic connections" depends on the value of the masking ratio, $\mathcal{R}_M$. We invite the reviewer to explore our interactive visualization at [this link](https://selfflowrebuttal.github.io/DTS_visualization/), which directly illustrates the effect of different masking ratio ($\mathcal{R}_M$) values on token masking patterns. The image and video experiments employ $\mathcal{R}_M = 0.25$ and $\mathcal{R}_M = 0.1$ respectively, where under $s \ll t$, the majority of tokens are heavily noised, fundamentally limiting the extent to which local interpolation can be exploited.
>
> Furthermore, the video generation results show very strong gains, with a margin of nearly 2 FVD points over REPA (Table 3), demonstrating the effectiveness of Self-Flow for domains with high redundancy.
>
> ### __Q3: Multi-modal joint training:__
> Please note that our experiments are conducted on a __wide-range of weights__, which were not manually tuned but automatically computed based on the data size and batch size per modality (see Appendix D). The considered weights include extreme weightings that favor each modality specifically (i.e., $w_I \gg \lbrace w_V, w_A \rbrace, w_V \gg \lbrace w_I, w_A \rbrace, w_A \gg \lbrace w_I, w_V \rbrace$) (L. 313 rightmost column - L. 370 leftmost column). The reference in the text for "we employ modality-specific loss weightings" just means that we test extensive variations where the weights across modalities are different. Fig. 6(a) interestingly shows that even with extreme weightings that favor each modality specifically, Self-Flow is still able to __improve generation quality across all modalities__ (L. 368-370). Furthermore, in Appendix D (specifically Fig. 15, Tab. 6), we enclose __all results for all considered weights after 1M multi-modal training steps__ (5 combinations), and show that all modalities improve under any weight selection, demonstrating robustness.
>
> We are happy to address any remaining questions.

---

> > ### Author Rebuttal · Reviewer_uWbc · 2026-04-07
> >
> > Thanks for your response. After carefully reading the comments, I will keep my original score.

---

### Decision · Program_Chairs · 2026-04-30

**Decision:**

Accept (regular)

**Comment:**

This paper received overall recommendations of 3, 4, 5, and 5. This paper introduces a self-supervised flow matching framework, which natively integrates representation learning into generative training via Dual-Timestep Scheduling to eliminate reliance on external encoders. The method achieves superior scalability and generalization across image, video, audio, and multi-modal generation tasks. During the rebuttal, the authors addressed the reviewers' concerns by providing a mathematical proof that aligns their objective with standard conditional flow matching, and by presenting strong 1B-parameter-scale results demonstrating significant compute efficiency over strong baselines like REPA. Although one reviewer maintained a weak reject score for spatial redundancy and ablation clarity, the authors' detailed interactive visualizations and extensive baseline comparisons sufficiently addressed these concerns. Given the impressive empirical performance and potential to influence scalable multi-modal generative modeling, this paper is recommended for acceptance.